# Zero Inflation as a Missing Data Problem: a Proxy-based Approach

**Trung Phung**[*1]    **Jaron J.R. Lee**[1]    **Opeyemi Oladapo-Shittu**[2]    **Eili Y. Klein**[2]    **Ayse Pinar Gurses**[1,2,3,4]    **Susan M. Hannum**[3]    **Kimberly Weems**[5,6]    **Jill A. Marsteller**[3,4]    **Sara E. Cosgrove**[2,4,5]    **Sara C. Keller**[2,4]    **Ilya Shpitser**[1]

[1]Johns Hopkins Whiting School of Engineering, Baltimore, MD
[2]Johns Hopkins University School of Medicine, Baltimore, MD
[3]Johns Hopkins Bloomberg School of Public Health, Baltimore, MD
[4]Johns Hopkins Medicine, Baltimore, MD
[5]Johns Hopkins Health System, Baltimore, MD
[6]Vassar Brothers Medical Center, Poughkeepsie, NY

## Abstract

A common type of zero-inflated data has certain true values incorrectly replaced by zeros due to data recording conventions (rare outcomes assumed to be absent) or details of data recording equipment (e.g. artificial zeros in gene expression data).

Existing methods for zero-inflated data either fit the observed data likelihood via parametric mixture models that explicitly represent excess zeros, or aim to replace excess zeros by imputed values. If the goal of the analysis relies on knowing true data realizations, a particular challenge with zero-inflated data is identifiability, since it is difficult to correctly determine which observed zeros are real and which are inflated.

This paper views zero-inflated data as a general type of missing data problem, where the observability indicator for a potentially censored variable is itself unobserved whenever a zero is recorded. We show that, without additional assumptions, target parameters involving a zero-inflated variable are not identified. However, if a proxy of the missingness indicator is observed, a modification of the effect restoration approach of Kuroki and Pearl allows identification and estimation, given the proxy-indicator relationship is known.

If this relationship is unknown, our approach yields a partial identification strategy for sensitivity analysis. Specifically, we show that only certain proxy-indicator relationships are compatible with the observed data distribution. We give an analytic bound for this relationship in cases with a categorical outcome, which is sharp in certain models. For more complex cases, sharp numerical bounds may be computed using methods in Duarte et al. [2023].

We illustrate our method via simulation studies and a data application on central line-associated bloodstream infections (CLABSIs).

[*]tphung1@jhu.edu

## 1  INTRODUCTION

Zero-inflated (ZI) data is prevalent in many empirical sciences such as public health, epidemiology, computational biology, and medical research. An important type of zero inflation occurs when some observed zeros of an outcome of interest do not represent true zero values.

As an example, consider patient surveillance for complications in outpatient settings, where any complication developed outside the hospital is of interest. One such complication is a central line-associated bloodstream infection (CLABSI) which can occur in patients undergoing therapies involving central venous catheters (CVCs). Such complications are fairly rare, but are associated with significant morbidity and mortality, and their prevalence is often assessed retrospectively. Because of this, absence of sufficient information on whether such a complication is present in a particular patient is often coded as a "presumed negative" rather than a "missing value" [Keller et al., 2020]. Since this type of value differs from a true negative value, indicating actual absence of a complication in a patient, the result is zero-inflated data. Another prominent example is single-cell RNA sequence data, whose zeros may signify either genuine values (representing, e.g. lack of gene expression) or artificial zeros resulting from technical artifacts of experimental protocols or recording equipment [Wagner et al., 2016, Jiang et al., 2022]. In all these cases, naive analysis of ZI data that does not distinguish true from artificial zeros can lead to markedly biased conclusions.

Existing approaches for zero inflation focus on observed data likelihood modelling using either hurdle models or zero-inflation models [Neelon et al., 2016, Greene, 2005]. Hurdles models are mixtures models of a distribution truncated at zero and another distribution modeling the occurrence of 0 values [Mullahy, 1986]. In genomics applications, Yu et al. [2023], Dai et al. [2023] use graphical models to represent the zero-inflated likelihood for the purposes of causal discovery. On the other hand, zero inflation models [Lambert, 1992, Young et al., 2022] assume two sources of

zeros, either structural (or inflated) zeros or true zeros due to sampling. More recent work has extended this type of approach to include semi-parametric models [Arab et al., 2012, Lam et al., 2006]. Kleinke and Reinecke [2013] apply an augmentation of the chained equations imputation approach to correct the bias introduced by inflated zeros. Lukusa et al. [2017] review methods in settings where inflated zeros co-occur with missing data, however these settings do not include cases considered here, where the excess zeros represent a censored realization.

The disadvantage of the first type of approach is that it does not aim to reconstruct underlying values, which are often of interest. The disadvantage of the second type of approach is that correctly distinguishing true from inflated zero values relies on assumptions that are unlikely to hold in practice, e.g., strict parametric assumptions. Moreover, these assumptions may not be congenial and not lead to a coherent full data distribution – guaranteeing model misspecification. This is a more general issue than zero inflation, and occurs in standard missing data problems as well. In contrast, our approach to modeling inflated zeros has two important features. First, we aim to distinguish true from inflated zeros, and thus identify underlying realizations in the data. Second, we avoid imposing strong parametric assumptions to do so.

Specifically, we propose to model zero inflation using a generalization of missing data models. In standard missing data, the relationship between an observed variable and its corresponding underlying variable is determined by an *observability indicator*. If the indicator is 1, the observed and the underlying variables coincide, while if the indicator is 0, the observed variable is recorded as a missing value. In zero inflated problems, we view improperly recorded zero values as missing values denoted by a zero. Hence, in this view, we cannot tell a zero indicating an actual value from a zero indicating missingness, and observing a zero means the observability indicator is *itself* unobserved.

This complication implies that even if we assume a missing data model where the full data distribution would have been identified absent zero inflation, such as the Missing-Completely-At-Random (MCAR) model, we would generally not obtain identification in the presence of zero inflation. Thus, the variant of the missing data problem we consider is significantly more complicated than standard missing data.

We approach this problem using recent theory of graphical models applied to missing data, which gives general identification results in the absence of zero inflation [Mohan et al., 2013, Bhattacharya et al., 2019, Nabi et al., 2020]. We first note that zero inflation problems viewed in this framework could be arranged in a hierarchy similarly to missing data problems [Rubin, 1976]: Zero-Inflated Missing-Completely-At-Random (ZI MCAR), Zero-Inflated Missing-At-Random (ZI MAR), and Zero-Inflated Missing-Not-At-Random (ZI MNAR).

We then show that if zero inflation is present, target parameters involving zero inflated variables are not identified without additional assumptions, even in the relatively simple ZI MCAR model. We further show that if an informative proxy for a missingness indicator exists, identification of the target parameters becomes possible provided the missing data model (sans zero inflation) is identified, via a modification of the effect restoration approach in Kuroki and Pearl [2014], provided the true proxy-indicator relationship is known.

If this relationship is not known, we show that only certain proxy-indicator relationships are compatible with the overall model which provides a natural sensitivity analysis strategy. In particular, in the case of a categorical outcome, we provide an analytic bound for the proxy-indicator relationship in the presence of zero inflation in a number of missing data models, and show that in some models our bound is sharp. In more general cases, we show that the numeric approach for obtaining bounds detailed in Duarte et al. [2023] may be used instead.

Finally, we demonstrate an application of our method on simulated data, as well as a real world dataset on CLABSIs.

## 2 GRAPHICAL MODELS OF MISSING DATA AND ZERO INFLATED DATA

In this section we briefly review relevant existing works on missing data, and describe difficulties posed by zero inflation.

### 2.1 MISSING DATA AND IDENTIFICATION

Let $X^{(1)} = \{X_1^{(1)}, \ldots, X_n^{(1)}\}$ be a set of random variables (r.v.s) of interest. Denote $\mathscr{X}_i^{(1)}$ as the state space of $X_i^{(1)}$, which we assume is categorical, and without loss of generality, includes the value 0. Samples of $X^{(1)}$ are systematically missing, with true values being replaced by a special symbol "?". To better represent missing data problems, it is convenient to use two additional sets of r.v.s: the proxies $X = \{X_1, \ldots, X_n\}$, where each proxy $X_i \in X$ has the state space $\mathscr{X}_i = \mathscr{X}_i^{(1)} \cup \{\text{"?"}\}$, and the binary observability indicators $R = \{R_1, \ldots, R_n\}$. Each proxy $X_i$ is deterministically defined in terms of the underlying variable $X_i^{(1)}$ and the observability indicator $R_i$ via the missing data version of the consistency rule: $X_i = X_i^{(1)}$ when $R_i = 1$ and $X_i = \text{"?"}$ when $R_i = 0$. Thus, a variable $X_i^{(1)}$ may be described as "$X_i$ had it (hypothetically) been observed", i.e., a counterfactual. The superscript notation is deliberately chosen to make the connection to counterfactuals in causal inference explicit. In addition to $X^{(1)}, R, X$, let $C$ represents other fully observed variables.

We define $R_{-i}$ as $\{R_1,\ldots,R_{i-1},R_{i+1},\ldots,R_n\}$, $R_{<i}$ as $\{R_1,\ldots,R_{i-1}\}$ and $R_{\geq i}$ as $\{R_i\ldots,R_n\}$, with analogous subsets of $X$, $X^{(1)}$ defined similarly. Following the nomenclature in Nabi et al. [2020], Bhattacharya et al. [2019], we call $p(X^{(1)},R,C)$ the full law, $p(R,X,C)$ the observed law, and $p(X^{(1)})$ the target law. A missing data model is a set of distributions over the variables $\{X^{(1)},R,X,C\}$ that satisfy the above consistency rule.

Following Mohan et al. [2013], we consider missing data model defined using a class of directed acyclic graphs (DAGs) called missing data DAGs (m-DAGs). Specifically, an m-DAG $\mathscr{G}(V)$ consists of nodes $V = \{X^{(1)},R,X,C\}$. Like all DAGs, m-DAGs only have directed edges and lack directed cycles, but also have a number of additional restrictions: each proxy $X_i$ has exactly 2 incoming edges $X_i^{(1)} \to X_i \leftarrow R_i$ (due to consistency); there is no edge from any $X_i$ or $R_i$ to any $X_i^{(1)}$. A joint $p(X^{(1)},R,X)$ in the missing data model corresponding to the m-DAG $\mathscr{G}$ factorizes as

$$\prod_{V \in \{R,X^{(1)}\}} p(V \mid \mathrm{pa}_{\mathscr{G}}(V)) \prod_{X_i \in X} p(X_i \mid R_i, X_i^{(1)})$$

where all terms $p(X_i \mid R_i, X_i^{(1)})$ are deterministic. Using m-DAGs, one can represent many interesting missing data scenarios, see Fig. 1 for examples.

An important goal in missing data problems, prior to statistical inference, is to ensure the target parameter, which is generally some function of the target law, is identified from the observed law. It follows by definition that the target law $p(X^{(1)})$ is identified if and only if the propensity score $p(R \mid X^{(1)})$ evaluated at $R = 1$ is identified, while the full law $p(X^{(1)},R)$ is identified if and only if the propensity score $p(R \mid X^{(1)})$ at all values of $R$ is identified. While identification of the target law is still an open problem, Nabi et al. [2020] showed a sound and complete method for identification of the full law $p(X^{(1)},R)$ from the observed law $p(R,X)$ in missing data models represented by m-DAGs and hidden variable m-DAGs.

## 2.2 ZERO INFLATION NON-IDENTIFIABILITY

A zero inflated (ZI) model associated with an m-DAG is a variant of the missing data model associated with that m-DAG, with the following important difference: the missing data consistency relating variables $X_i^{(1)} \in X^{(1)}, X_i \in X, R_i \in R$ is replaced by a zero inflation version, where $X_i = X_i^{(1)}$ if $R_i = 1$, and $X_i = 0$ if $R_i = 0$. [1]

There are several important consequences of zero inflated consistency. Firstly, both $X_i^{(1)} \in X^{(1)}$ and $X_i \in X$ take values in $\mathscr{X}_i$, and no variable in a ZI problems takes the value "?". Secondly, as in missing data, the ZI-variable $X_i^{(1)}$ is coun-

---

[1]Note that we consider ZI models with categorical state spaces only, unless stated otherwise.

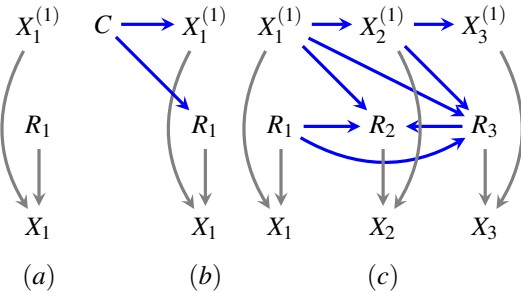

Figure 1: Missing data scenarios represented by m-DAG. Circle nodes denote observed variables, while others nodes are unobserved. Gray edges denote deterministic nature of $p(X_i \mid R_i, X_i^{(1)})$ due to consistency. (a) $X_1^{(1)}$ is MCAR since $R_1 \perp\!\!\!\perp X^{(1)}$. (b) $X_1^{(1)}$ is MAR since $R_1 \perp\!\!\!\perp X^{(1)} \mid C$. (c) $X_1^{(1)}, X_2^{(1)}, X_3^{(1)}$ are MNAR, since observability indicators $R_1, R_2, R_3$ are are not independent of these variables, either marginally or given observed variables.

terfactual, and according to the ZI consistency rule, its true realizations are observed only when $R_i = 1$. In particular, if $X_i = x \neq 0$, we deduce $R_i = 1$ and $X_i^{(1)} = x$. However, since it is not possible to tell whether a realization $X_i = 0$ corresponds to the situation where 0 is the true value of $X_i^{(1)}$, or corresponds to a censored realization of $X_i^{(1)}$, $R_i$ is *unobserved* whenever $X_i = 0$. Moreover, while we still refer to $p(X^{(1)})$ and $p(X^{(1)},R)$ as the target law and the full law, respectively, we will refer to $p(R,X)$ as the zero-inflated law (ZI law), rather than the observed law, since $R$ is not always observed. Thirdly, the ZI consistency imposes the following important restriction on the ZI law $p(R,X)$

**(Z)** For every $i$ and $x \neq 0$, $p(R_i = 0, X_i = x) = 0$.

We classify ZI models as ZI MCAR, ZI MAR, or ZI MNAR, if its missing data version is MCAR, MAR, or MNAR, respectively. Examples of ZI models are shown in Fig. 2 and Fig. 3.

Just as in missing data problems, the goal in ZI problems is to identify (a function of) the target law or the full law from the observed law and possibly additional objects. We focus on the full law identification in this paper. Unsurprisingly, ZI problems are significantly harder than missing data problems, in the sense that both the target law and the full law are non-parametrically non-identified even in the simplest setting (ZI MCAR), as shown by the following result.

**Lemma 1** (Non-identifiability). *Given a ZI model associated with any m-DAG $\mathscr{G}$, both the target law $p(X^{(1)})$ and the full law $p(X^{(1)},R,C)$ are non-parametrically non-identified.*

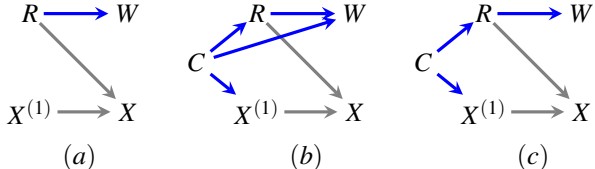

Figure 2: Examples of proxy-augmented ZI MCAR model (a) and ZI MAR models (b and c). **A1**, **A2** holds in (a), **A1**[†], **A2**[†] hold in (b), and **A1**[*], **A2**[*] hold in (c). Unlike missing data, indicator $R$ is partially observed.

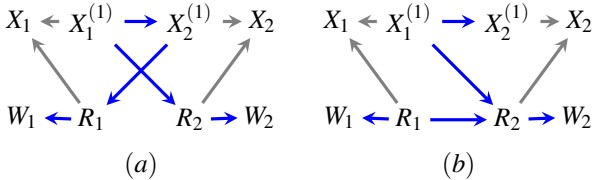

Figure 3: Examples of proxy-augmented ZI MNAR models. (a) ZI bivariate block-parallel model. (b) ZI bivariate block-sequential MAR model.

# 3 PROXY-BASED IDENTIFICATION

We first demonstrate our approach to proxy-based identification with the simplest ZI missing data model, ZI MCAR, and generalize it to arbitrary ZI m-DAG.

## 3.1 IDENTIFICATION IN THE ZI MCAR MODEL

Lemma 1 implies that any identification method must rely on additional assumptions beyond those implied by the m-DAG. To illustrate additional assumptions that will be employed, consider a simple ZI MCAR model with a single ZI variable $X^{(1)}$ taking values in $\{0, \dots, K\}$, and the corresponding inflation indicator $R$, where $R \perp\!\!\!\perp X^{(1)}$.

To simplify subsequent presentation, we will use the following notational shorthand: $p_{a_i|b_j}$ to mean $p(A = i \mid B = j)$, and $\mathbf{p}_{A|B}$ to mean the stochastic matrix whose elements are $p_{a_i|b_j}$. Similarly for $p_{a_i,b_j}$ and $p_{A,B}$. We also use a matrix multiplication shorthand, where $p_{a_i|b_j} p_{b_j|c_k} = p_{a_i|c_k}$ is taken to mean $\sum_j p(A = i|B = j)p(B = j|C = k) = p(A = i|C = k)$.

We will assume the existence of an observed binary proxy variable $W$ informative for $R$ with the following properties:

**(A1)** $W \perp\!\!\!\perp X^{(1)} \mid R$,

**(A2)** The matrix $\mathbf{p}_{W|R}$ is invertible.

Note that since $W$ and $R$ are binary, **A2** is equivalent to $p_{w_0|r_0} \neq p_{w_0|r_1}$. Due to the existence of the proxy variable $W$, we call this ZI MCAR model "proxy-augmented", whose graph is shown in Fig. 2 (a).

Kuroki and Pearl [2014] considered assumptions **A1** and **A2** in the context of obtaining identification of causal effects in the presence of unobserved confounding. In that work, the proxy variable $W$ was related to an unobserved categorical variable which was a common cause of the treatment and outcome variables.

In this paper, we adopt the method of Kuroki and Pearl [2014] to express the ZI law $p(R, X)$ in terms of the observed law $p(X, W)$ and the conditional distribution $p(W \mid R)$. In addition to **A1** and **A2**, the Kuroki-Pearl method requires that the observed law $p(X, W)$ and $p(W \mid R)$ are from the same full law (e.g. compatible), and $p(W \mid R)$ is known.

To see that point identification is then possible, we write $p(wx) = \sum_r p(w \mid r)p(r, x)$ in matrix form

$$\underbrace{\begin{pmatrix} p_{w_0,x_0} & \cdots & p_{w_0,x_K} \\ p_{w_1,x_0} & \cdots & p_{w_1,x_K} \end{pmatrix}}_{\mathbf{p}_{WX}} = \underbrace{\begin{pmatrix} p_{w_0|r_0} & p_{w_0|r_1} \\ p_{w_1|r_0} & p_{w_1|r_1} \end{pmatrix}}_{\mathbf{p}_{W|R}} \underbrace{\begin{pmatrix} p_{r_0,x_0} & \cdots & 0 \\ p_{r_1,x_0} & \cdots & p_{r_1,x_K} \end{pmatrix}}_{\mathbf{p}_{RX}},$$

$$(1)$$

where the 0 entry in $\mathbf{p}_{RX}$ is due to the restriction **Z**. Since $\mathbf{p}_{W|R}$ is invertible, we can solve for $\mathbf{p}_{RX}$ by $[\mathbf{p}_{W|R}]^{-1}\mathbf{p}_{WX}$, leading to the following result.

**Theorem 1** (ZI law restoration in ZI MCAR). *For the ZI MCAR model in Fig. 2 (a) under* **A1**, **A2**, *the ZI law $p(R, X, W)$ is point identified given the observed law $p(X, W)$ and a compatible proxy-indicator conditional distribution $p(W \mid R)$, as follows*

$$p(r, x, w) = p(w \mid r) \left[ \mathbf{p}_{W|R}^{-1} \mathbf{p}_{WX} \right]_{r,x}. \quad (2)$$

After the ZI law $p(R, X, W)$ is identified, the full law is identified, $p(X^{(1)}, R, W) = p(X, W \mid R = 1)p(R)$, by standard assumptions of the MCAR model.

**Remark.** *There are two difficulties with this result. First, since $R$ is potentially unobserved, it is not always reasonable to specify the true $p(W \mid R)$ in applications. Second, since our working model corresponds to a hidden variable DAG, the model imposes restrictions on the pair $(p(X, W), p(W \mid R))$, meaning that not every potential distribution $p(W \mid R)$ would be consistent with the observed data law under our model. Using inconsistent $p(W \mid R)$ in the matrix inversion equation places us outside the model, and can yield inconsistent results, such as invalid negative probabilities $p(R, X)$. Examples of such an inconsistency is provided in the Appendix. Kuroki and Pearl [2014] noted the latter issue in the context of causal inference, but did not provide bounds.*

## 3.2 PROXY-BASED IDENTIFICATION IN GENERAL ZI MISSING DATA MODELS

In this section, we generalize our previous proxy-based approach to an arbitrary graphical ZI model corresponding

to an m-DAG, given that the full law is point identified in the missing data model associated to that m-DAG.

Consider any ZI model associated with an arbitrary m-DAG, with a set of fully observed covariates $C$, a set of ZI variables $X^{(1)} = \{X_1^{(1)}, \ldots, X_n^{(1)}\}$, inflation indicators $R = \{R_1, \ldots, R_n\}$, observed versions $X = \{X_1, \ldots, X_n\}$ for variables in $X^{(1)}$, and proxies $W = \{W_1, \ldots, W_n\}$ for variables in $R$.

We make the following assumptions which generalize **A1** and **A2**:

(**A1**$^*$) $\forall i$, $W_i \perp\!\!\!\perp X^{(1)}, C, R_{-i} \mid R_i$.

(**A2**$^*$) The matrix $\mathbf{p}_{W|R}$ is invertible.

In addition, we will provide alternatives to **A1**$^*$ and **A2**$^*$ which allow the proxies $W$ to potentially depend on $C$:

(**A1**$^\dagger$) $\forall i$, $W_i \perp\!\!\!\perp X^{(1)}, R_{-i} \mid C, R_i$.

(**A2**$^\dagger$) The matrix $\mathbf{p}_{W|R,c}$ is invertible for every value $c$.

The identification strategy we adopt proceeds in two stages:

1. **ZI law restoration**: point identify (if true $p(W \mid R)$ is known) or partially identify the ZI law $p(R, X, W, C)$ from the observed law $p(X, W, C)$.

2. **Downstream identification**: identify the full law $p(X^{(1)}, R, W, C)$ from the ZI law $p(R, X, W, C)$.

Since every $R_i$ is unobserved whenever $X_i = 0$ in ZI problems, the purpose of the first stage is to recover the ZI law involving $R$ and other observed variables. Under mentioned proxy assumptions and knowledge of the true $p(W \mid R)$, point identification of this law is possible. Otherwise, partial identification bounds are computed. If point or partial identification is possible, variables in $R$ may now be treated as observed data, and the problem is reduced to classical identification in missing data model. In particular, we adopt the sound and complete identification procedure described by Nabi et al. [2020] to point identify the full law in the second stage.

While we focus on non-parametric point identification results for the full law, one could instead employ any point or partial identification procedure developed for missing data problems for the second stage. We leave these types of extensions to future work.

### 3.2.1 ZI law restoration

Under the proxy assumptions, we have the following identification result, which generalizes Theorem 1.

**Theorem 2** (ZI law restoration). *Given a ZI model satisfying assumptions* **A1**$^*$ *and* **A2**$^*$ *(or* **A1**$^\dagger$ *and* **A2**$^\dagger$*), the ZI law $p(R, X, W, C)$ is point identified given the observed law*

$p(X, W, C)$ *and a compatible proxy-indicator conditional distribution $p(W \mid R)$ (OR $p(W \mid R, C)$),*

$$p(r, x, w, c) = p(w \mid r)\left[\mathbf{p}_{W|R}^{-1}\mathbf{p}_{WXC}\right]_{r,x,c} under \ \mathbf{A1}^*, \mathbf{A2}^*$$
$$p(r, x, w, c) = p(w \mid r, c)\left[\mathbf{p}_{W|R,C}^{-1}\mathbf{p}_{WXC}\right]_{r,x,c} under \ \mathbf{A1}^\dagger, \mathbf{A2}^\dagger.$$
(3)

Theorem 2 suffers from the same issue as Theorem 1: it is unlikely that the true distribution $p(W \mid R)$ (or $p(W \mid R, C)$) will always be available, and given a candidate distribution, it is not obvious to verify that it is compatible with the model and the observed law.

If the true $p(W \mid R)$ (or $p(W \mid R, C)$) is not given, we must find the set of compatible $p(W \mid R)$ (or $p(W \mid R, C)$) distributions to the model and the observed law. In general, bounds on $p(W \mid R)$ (or $p(W \mid R, C)$) may be computed numerically by encoding the model as a system of polynomial equations and finding extrema of this system using polynomial programming. A method for solving such systems of equations using a primal/dual method is described in Duarte et al. [2023]. These bounds lead to a natural sensitivity analysis strategy according to our two stage approach. Particularly, each compatible $p(W \mid R)$ in the bounds implies a valid ZI law by Theorem 2, which in turn implies a full law by Proposition 1 of the next section. In Section 4, we conduct a grid search of the compatible set to illustrate this point.

While numeric bound computation is a general approach, finding such bounds is computationally challenging due to the need to solve polynomial programs. Fortunately, we show that in certain ZI models, it is possible to derive analytic bounds on $p(W \mid R)$ (or $p(W \mid R, C)$), instead. We also show that these bounds are sharp in some cases.

### 3.2.2 Downstream identification

After the ZI law $p(R, X, W, C)$ is recovered in the restoration step, one may consider this law as the "observed law" in the missing data problem corresponding to the same m-DAG, and invoke missing data identification to obtain the full law $p(X^{(1)}, R, W, C)$. We note that this second identification stage is not precisely the same as that for standard missing data problems, because identification relies on consistency, and consistency under ZI differs from missing data consistency whenever $R = 0$.

Fortunately, consistency when $R = 1$ coincides in ZI problems and missing data problems, and, as the following result shows, suffices for identification.

**Proposition 1** (ZI full law identification). *The full law $p\left(X^{(1)}, R, W, C\right)$ exhibiting zero inflation that is Markov relative to an m-DAG $\mathcal{G}$ is identified given the ZI law $p(R, X, W, C)$ if and only if $\mathcal{G}$ does not contain edges of*

*the form $X_i^{(1)} \to R_i$ (no self-censoring) and structures of the form $X_j^{(1)} \to R_i \leftarrow R_j$ (no colliders), and the positivity assumption holds. Moreover, the identifying functional for the full law coincides with the functional given in Malinsky et al. [2021].*

## 3.3 PARTIAL IDENTIFICATION IN ZI MCAR

In this subsection, we relax the requirement that the true $p(W \mid R)$ must be given in the ZI law restoration step, and provide bounds for this conditional distribution in the proxy-augmented ZI MCAR model.

Consider the proxy-augmented ZI MCAR model in Fig. 2 (a). This model is equivalently described by the following model $\mathscr{P}$, satisfying **Z**, **A1**, and **A2**:

$$\mathscr{P} = \begin{cases} (\mathbf{q}_{W|R}, \mathbf{q}_{RX}): & \mathbf{q}_{W|R} \geq 0, \sum_w q_{w|r} = 1, \forall r, \\ & \mathbf{q}_{RX} \geq 0, \sum_{rx} q_{rx} = 1, \\ & \forall x \neq 0 (q_{r_0 x} = 0); q_{w_0|r_0} \neq q_{w_0|r_1}. \end{cases}$$
(4)

Given an observed law $p(X, W)$, we are interested in the following subset $\mathscr{Q} \subseteq \mathscr{P}$ of distributions yielding the observed law,

$$\mathscr{Q} = \left\{ (\mathbf{q}_{W|R}, \mathbf{q}_{RX}) \in \mathscr{P} : \mathbf{q}_{W|R} \mathbf{q}_{RX} = \mathbf{p}_{WX} \right\}.$$
(5)

In particular, our goal is finding all $\mathbf{q}_{W|R} \in \mathscr{Q}$, which is the **partial identification** of $q(W \mid R)$ w.r.t. the given observed law. This is equivalent to projecting $\mathscr{Q}$ onto the probability simplex of $\mathbf{q}_{W|R}$. From (5) and (4), one way to check whether an invertible $\mathbf{q}_{W|R} \in \mathscr{Q}$ is to compute $\mathbf{q}_{RX} = (\mathbf{q}_{W|R})^{-1} \mathbf{p}_{WX}$ and check $(\mathbf{q}_{W|R}, \mathbf{q}_{RX}) \in \mathscr{Q}$. First, $\mathbf{q}_{RX}$ must be a stochastic matrix for any problem under **A1** and **A2**. Second, $\mathbf{q}_{RX}$ must also satisfy ZI-consistency constraint **Z**. If these conditions are true, there is a joint distribution in the model generates both $\mathbf{q}_{W|R}$ and $\mathbf{p}_{WX}$, and they are said to be **compatible**. After the compatible set of $\mathbf{q}_{W|R}$ is derived, the partial identification of $\mathbf{q}_{RX}$ could be obtained using (2).

We note that **Z** implies, for all $x \neq 0$, $q_x = q_{r_1,x}$, so $q_{r_1|x} = 1$. Then by considering $q_{w_0|r_0} q_{r_0,x} + q_{w_0|r_1} q_{r_1,x} = p_{w_0,x}$, we obtain point identification $q_{w_0|r_1} = p_{w_0|x_1}$ and the marginal constraints $\forall x \neq 0, p_{w_0|x} = p_{w_0|x_1}$. Note that these constraints may be used to design a falsification test of the model.

However, $q_{w_0|r_0}$ is not identified, and its bounds must be obtained by solving the following polynomial program:

$$\max_{q_{w_0|r_0}} \quad \pm q_{w_0|r_0}$$
$$\text{s.t.} \quad \mathbf{q}_{W|R} \mathbf{q}_{RX} = \mathbf{p}_{WX},$$
$$\mathbf{q}_{W|R} \geq 0, \forall r (\sum_w q_{w|r} = 1), q_{w_0|r_0} \neq q_{w_0|r_1},$$
$$\mathbf{q}_{RX} \geq 0, \sum_{rx} q_{rx} = 1, q_{w_0|r_1} = p_{w_0|x_1}.$$
(6)

Since both $q_{w_0|r_0}$ and $\mathbf{q}_{RX}$ are unknowns, the above system of equations corresponds to a quadratic program, which is difficult to solve in general.

However, it is possible to transform this optimization into an equivalent linear program with the following observations:

1. A specific solution to $\mathbf{q}_{RX}$ is not required. One merely needs to check if $\mathbf{q}_{W|R}^{-1} \mathbf{p}_{WX}$ is a stochastic matrix.

2. If $\mathbf{q}_{W|R} \mathbf{q}_{RX} = \mathbf{p}_{WX}$, where all matrices are non-negative, $\mathbf{p}_{WX}$ sum to 1 and $\mathbf{q}_{W|R}$ sum to 1, then $\mathbf{q}_{RX}$ sum to 1. The proof of this fact is in the Appendix.

3. The inverse $\left[ \mathbf{q}_{W|R} \right]^{-1}$ is

$$\frac{1}{q_{w_0|r_0} - q_{w_0|r_1}} \begin{pmatrix} 1 - q_{w_0|r_1} & -q_{w_0|r_1} \\ q_{w_0|r_0} - 1 & q_{w_0|r_0} \end{pmatrix}$$
(7)

Observations 1 and 2 imply that checking compatibility involves only checking non-negativity of $\mathbf{q}_{W|R}^{-1} \mathbf{p}_{WX}$, reducing the unknowns in our optimization problem to only $q_{w_0|r_0}$. Checking $\mathbf{q}_{W|R}^{-1} \mathbf{p}_{WX}$ is still non-linear in $\mathbf{q}_{W|R}$, but (7) suggests an equivalent procedure consisting of two separate problems where $q_{w_0|r_0} > q_{w_0|r_1}$ or $q_{w_0|r_0} < q_{w_0|r_1}$, respectively. Concretely, for each case $s = 1$ and $s = -1$, we consider 2 linear programs

$$\max_{q_{w_0|r_0}} \quad \pm q_{w_0|r_0}$$
$$\text{s.t.} \quad s \cdot \begin{pmatrix} 1 - q_{w_0|r_1} & -q_{w_0|r_1} \\ q_{w_0|r_0} - 1 & q_{w_0|r_0} \end{pmatrix} \mathbf{p}_{WX} \geq \mathbf{0},$$
$$s \cdot q_{w_0|r_0} > s \cdot q_{w_0|r_1}, 0 \leq q_{w_0|r_0} \leq 1,$$
$$q_{w_0|r_1} = p_{w_0|x_1}.$$
(8)

These problems could be solved analytically using fast linear program solvers, yielding the following partial identification result for $p(W \mid R)$. A detailed proof is in the Appendix.

**Theorem 3** (ZI MCAR compatibility bound). *Consider a ZI MCAR model in Fig. 2 (a) under proxy assumptions **A1**, **A2**, with categorical $X$ and binary $R, W$. Given a consistent observed law $p(X, W)$ satisfying positivity assumption, the set of compatible proxy-indicator conditionals $q(W \mid R)$ is given by*

$$q_{w_0|r_1} = p_{w_0|x_1}$$

$$q_{w_0|r_0} \in \begin{cases} [p_{w_0|x_0}, 1] & \text{if } p_{w_0|x_0} > p_{w_0|x_1} \\ [0, p_{w_0|x_0}] & \text{if } p_{w_0|x_0} < p_{w_0|x_1} \\ (0, 1) \setminus \{ p_{w_0|x_0} \} & \text{if } p_{w_0|x_0} = p_{w_0|x_1} \end{cases}$$

*These bounds are sharp. Moreover, if $p_{w_0|x_0} = p_{w_0|x_1}$, $p(X, W)$ must satisfy $0 < p_{w_0|x_1} < 1$, and zero inflation does not occur, i.e., $q(R = 0) = 0$.*

## 3.4 PARTIAL IDENTIFICATION IN ZI MAR

We compute analytical bounds for two versions of the proxy-augmented ZI MAR model, illustrated in Fig. 2 (b) and (c). The first model has $C \to W$ and satisfies $\mathbf{A1}^\dagger$ and $\mathbf{A2}^\dagger$, while $C \not\to W$ in the second model, and the proxy assumptions are $\mathbf{A1}^*$ and $\mathbf{A2}^*$.

In the first proxy-augmented ZI MAR model, the set of compatible $p_{W|R,C}$ is given by the Cartesian product of the independently determined ZI MCAR bounds for each value $c$. This leads to the following direct analogue of Theorem 3. The proof is deferred to the Appendix.

**Theorem 4** (ZI MAR compatibility bound 1). *Consider a ZI MAR model in Fig. 2 (b) under proxy assumptions $\mathbf{A1}^\dagger$ and $\mathbf{A2}^\dagger$, with categorical $X, C$ and binary $R, W$. Given a consistent observed law $p(X, W, C)$ satisfying positivity, the set of compatible proxy-indicator conditional distributions $q(W \mid R, C)$ is given by, for each value $c$,*

$$q_{w_0|r_1,c} = p_{w_0|x_1,c}$$

$$q_{w_0|r_0,c} \in \begin{cases} [p_{w_0|x_0,c}, 1] & \text{if } p_{w_0|x_0,c} > p_{w_0|x_1,c} \\ [0, p_{w_0|x_0,c}] & \text{if } p_{w_0|x_0,c} < p_{w_0|x_1,c} \\ (0,1) \setminus \{p_{w_0,|x_0,c}\} & \text{if } p_{w_0|x_0,c} = p_{w_0|x_1,c} \end{cases}$$

*These bounds are sharp. Moreover, if $p_{w_0|x_0,c} = p_{w_0|x_1,c}$, $p(X, W, C)$ must satisfy $0 < p_{w_0|x_0,c} < 1$, and zero inflation does not occur for stratum $C = c$, i.e., $q(R = 0 \mid c) = 0$.*

On the other hand, the compatibility bound for the second ZI MAR model is the intersection of the ZI MCAR bounds for each values $c$. The proof is deferred to the Appendix.

**Theorem 5** (ZI MAR compatibility bound 2). *Consider a ZI MAR model in Fig. 2 (c) under proxy assumptions $\mathbf{A1}^*$ and $\mathbf{A2}^*$, with categorical $X, C$ and binary $R, W$. Given a consistent observed law $p(X, W, C)$ satisfying positivity, the set of compatible proxy-indicator conditional distributions $q(W \mid R)$ is given by*

$$q_{w_0|r_1} = p_{w_0|x_1}$$

$$q_{w_0|r_0} \in \begin{cases} [\max_c p_{w_0|x_0,c}, 1] & \text{if } \exists \tilde{c}, p_{w_0|x_0,\tilde{c}} > p_{w_0|x_1}, \\ [0, \min_c p_{w_0|x_0,c}] & \text{if } \exists \tilde{c}, p_{w_0|x_0,\tilde{c}} < p_{w_0|x_1}, \\ (0,1) \setminus \{p_{w_0|x_1}\} & \text{if } \forall c, p_{w_0|x_0,c} = p_{w_0|x_1}. \end{cases}$$

*These bounds are sharp. Moreover, if $\forall c, p_{w_0|x_0,c} = p_{w_0|x_1}$, $p(X, W, C)$ must satisfy $\forall c, 0 < p_{w_0|x_0,c} < 1$, and zero inflation does not occur, i.e., $q(R = 0) = 0$.*

Note that the first two cases are mutually exclusive due to the following lemma.

**Lemma 2.** *For a ZI MAR model in Fig. 2 (b) under $\mathbf{A1}^\dagger$ and $\mathbf{A2}^\dagger$, the observed law $p(X, W, C)$ obeys*

$$\forall c, \forall x \neq 0, p_{w_0|x,c} = p_{w_0|x_1,c}. \tag{9}$$

*For a ZI MAR model in Fig. 2 (c) under $\mathbf{A1}^*$ and $\mathbf{A2}^*$, the observed law $p(X, W, C)$ obeys*

$$\forall c, \forall x \neq 0, p_{w_0|x,c} = p_{w_0|x_1}, \tag{10}$$

$$\text{either } \forall c \left( p_{w_0|x_0,c} \leq p_{w_0|x_1} \right) \text{ or } \forall c \left( p_{w_0|x_0,c} \geq p_{w_0|x_1} \right).$$

Note that, as before, the marginal constraints described may be used to design a model falsification test.

## 3.5 PARTIAL IDENTIFICATION IN ZI MNAR

Consider the ZI version of any MNAR model represented by an m-DAG where the target law is identified. In missing data, an important subclass of such models are submodels of the no-self-censoring model in Malinsky et al. [2021] due to the results in Nabi et al. [2020]. The ZI versions of such models exhibit a crucial complication not found in previously discussed ZI models, namely that multiple variables may be zero inflated. For these models, we posit a set of proxies $W = \{W_1, \ldots, W_n\}$ corresponding to $R = \{R_1, \ldots, R_n\}$, and assume assumptions $\mathbf{A1}^*$, $\mathbf{A2}^*$ in Section 3.2.1 are satisfied. Fig. 3 (a) and (b) show two bivariate examples of such models. We use the short hand $p_{w_{ka}|x_{kb}} = p(W_k = a \mid X_k = b)$.

Given observed law $p(X, W, C)$, we seek the compatible set of $\{q(W_k \mid R_k)\}_{k=1}^n$, whose elements allow restoration of $p(R, X, W, C)$ via Theorem 2. Although sharp bounds for $\{q(W_k \mid R_k)\}_{k=1}^n$ are unknown, the ZI MAR partial identification procedure could be applied to each $R_k$ independently to obtain bounds for $q_{w_{k0}|r_{k0}}$. Moreover, due to the usual properties of ZI, $q_{w_{k0}|r_{k1}}$ is point identified for each $k$.

For each $k$, we apply Theorem 5 with $X_k, R_k, W_k$ being $X, R, W$, respectively, and $Z_k \triangleq \{X, W, C\} \setminus \{X_k, W_k\}$ being the covariates $C$. These bounds are not sharp as structural constraints of the MNAR model are not considered. However, these bounds are valid in the sense that the Cartesian product of these bounds contains the true model compatible set of distributions $\{p(W_k \mid R_k) : k = 1, \ldots n\}$.

In addition, we note that (11) below hold in the observed law under our model, and may be used as falsification test for our ZI model.

**Lemma 3.** *Consider any ZI model in Section 3.2.1 under $\mathbf{A1}^*$ and $\mathbf{A2}^*$. Denote $Z_k \triangleq \{X, W, C\} \setminus \{W_k, X_k\}$. The observed law $p(X, W, C)$ must satisfy, for each $k$,*

$$\forall z_k, \forall x \neq 0 \, p_{w_{k0}|x_k=x,z_k} = p_{w_{k0}|z_{k1}}, \tag{11}$$

$$\forall z_k \left( p_{w_{k0}|x_{k0},z_k} \leq p_{w_{k0}|x_{k1}} \right) \text{ or } \forall z_k \left( p_{w_{k0}|x_{k0},z_k} \geq p_{w_{k0}|z_{k1}} \right).$$

## 3.6 IDENTIFICATION GIVEN A KNOWN ZERO INFLATION PROBABILITY

For ZI MCAR models in Theorem 3 and ZI MAR model in Theorem 5, we provided the identification $q_{w_0|r_1} = p_{w_0|x_1}$

and the bounds for $q_{w_0|r_0}$, which lead to partial identification of the full law $p(X^{(1)}, R, X, W, C)$.

If $q_{w_0|r_0}$ is known a priori, the full law is point identified. Alternatively, point identification of the full law may be obtained if the zero inflation probability, or $p(R = 0)$, is known.

This is because the joint distribution $p(W, R)$ for binary $W, R$ has dimension 3, and one (variationally dependent) parameterization for this joint is via the following 3 parameters $p(R = 0), p(W = 0), q_{w_0|r_1}$. This is easy to see by noting that we can compute $p(R = 0, W = 0) = q_{w_0|r_1}(1 - p(R = 0))$, and $p(R = 0)$, $p(W = 0)$, and $p(R = 0, W = 0)$ are the Möbius parameters for $p(W, R)$ [Evans and Richardson, 2014].

In particular, we have the following: $q_{w_0|r_0} = \frac{p_{w_0} - q_{w_0|r_1} p_{r_1}}{p_{r_0}} = \frac{p_{w_0} - p_{w_0|x_1} p_{r_1}}{p_{r_0}}$, which in turns implies point identification of the full law.

Note that not every zero inflation probability $p(R = 0)$ is compatible with the model. This is easily seen by noting that the Möbius parameterization is variationally dependent, and two parameters, namely $p(W = 0)$ and $q_{w_0|r_1}$, are known. Howevre, our derived bounds for $q_{w_0|r_1}$ naturally imply bounds for $p(R = 0)$, with sharp bounds for the former implying sharp bounds for the latter.

# 4 EXPERIMENTS

We confirmed the validity of our analytical results for inflated zero models by sampling data generating processes (DGPs), and numerical methods. In addition, we used our methods to perform sensitivity analyses on CLABSI data. Details of these experiments are in the Appendix.

## 4.1 BOUND VALIDITY IN RANDOM DGPS

We verify the results of Theorem 3, Theorem 5 and related observed law constraints by randomly generating DGPs in models we described. We generated $10^8$ DGPs in the model in Fig. 2 (a), satisfying ZI-consistency, **A1**, **A2**, and $10^8$ DGPs in the model in Fig. 2 (b), satisfying ZI-consistency, **A1**\*, **A2**\*. For both cases, we verified identification of $q_{w_0|r_1}$ and the bounds for $q_{w_0|r_0}$ as predicted by the corresponding theorem. For the bounds, two tests were conducted

1. *Bound validity*: is the true $p_{w_0|r_0}$ inside the bounds?
2. *Model consistency*: grid search the bound, compute $p(r, x)$ (or $p(r, x, c)$) according to Theorem 1 (or 2), and verify that these are probability distributions.

Additionally, for ZI MAR, we checked marginal constraints in (10). We found that all considered results held up to floating point precision in every single DGP.

| MCAR | lb | ub | num lb | num ub | $p_{w_0|r_0}$ |
|---|---|---|---|---|---|
| 0 | 0.5564 | 1 | 0.5564 | 1 | 0.8207 |
| 1 | 0.3578 | 1 | 0.3578 | 1 | 0.4936 |
| 2 | 0 | 0.5206 | 0 | 0.5206 | 0.4536 |
| 3 | 0.6064 | 1 | 0.6064 | 1 | 0.6826 |
| MAR | lb | ub | num lb | num ub | $p_{w_0|r_0}$ |
| 0 | 0 | 0.4290 | 0 | 0.4290 | 0.4132 |
| 1 | 0.8346 | 1 | 0.8346 | 1 | 0.8486 |
| 2 | 0 | 0.3404 | 0 | 0.3404 | 0.3192 |
| 3 | 0.3002 | 1 | 0.3002 | 1 | 0.5155 |

Table 1: Comparison between our analytical lower and upper bound (*lb/ub*) to numerical bounds (*num lb/num ub*) for a randomly selected set of DGPs. True $p_{w_0|r_0}$ is reported.

## 4.2 BOUNDS BY NUMERICAL METHODS

We compared our analytical bounds with numerical bounds computed using `autobounds` package in Duarte et al. [2023] for a subset of DGPs used for verification of bound validity in Section 4.1.

In particular, 20 DGPs were randomly selected for each model (ZI MCAR and ZI MAR), and their observed laws were computed. For each DGP, 2 polynomial programs were constructed, whose objective functions are maximizing or minimizing $p(W = 0|R = 0)$, respectively, and whose constraints are (i) structural constraints from the corresponding graph, (ii) probability constraints, (iii) ZI-consistency constraint, (iv) constraints resulted from the structure imposed on the observed law by the structure of the full law. The solutions to these programs are the numerical lower and upper bounds of $p(W = 0|R = 0)$. We refer reader to Duarte et al. [2023] for details of the program's construction and the methods used by the polynomial program solver.

For all DGPs, the numerical bounds coincided with our analytical bounds up to the 4th decimal place. Since the algorithm in the `autobounds` package is an anytime algorithm, our analytic bounds were always contained inside the numerical bounds. Table 1 shows a selection of these results.

## 4.3 DATA APPLICATION

Patients receiving therapies involving central venous catheters (CVCs) through home infusion agencies may develop CLABSI. Though relatively rare, CLABSIs are potentially dangerous. Knowing true CLABSI rates is essential in deploying and testing the impact of CLABSI prevention activities. Recorded CLABSI rates undercount true positive cases. This is because adjudicators performing CLABSI surveillance often lack access to the full information required to determine whether a CLABSI has occurred [Hannum et al., 2022, 2023]. If the available information do not

meet the CLABSI definition criteria, as CLABSIs are relatively rare, the adjudicator typically records the CLABSI status as a presumed negative.

We will apply our zero inflation correction method to data on patients undergoing CVC therapies and thus potentially susceptible to a CLABSI. Our data contains 652 unique patient records obtained from five different home infusion agencies across 14 states and the District of Columbia, see [Keller et al., 2023] for additional details. These records correspond to records investigated on patients who presented to a hospital due to a complication and on whom blood cultures were drawn and were positive. Many patients with CVCs who presented to the hospital due to a complication on whom blood cultures were drawn and were positive do have CLABSIs. In fact, the observed CLABSI rate in our data was more than 65%, much higher than the prevalence in the population undergoing CVC therapies. However due to zero inflation, even the elevated observed CLABSI rate undercounts the true CLABSI rate in this cohort.

Variables in our data included covariates $C$, which indicated home infusion therapy type and CVC type, coded as binary variables. A description of these covariates is found in the Appendix. The outcome of interest is the true CLABSI probability (had zero inflation not occurred), which we denote by $X^{(1)}$. This outcome is not directly observed. Instead, our data contains the observed CLABSI status $X$, recorded as 0 and 1. Given this variable, we define the inflation indicator $R$ which corresponds to the adjudicator having enough information to make a CLABSI determination for a particular case. The information could come from private meeting with patients and specialists, or from reading patients test results and other data in health record systems. Recording conventions dictate that this indicator has a known value whenever the observed CLABSI is 1, and is unobserved otherwise (since we cannot distinguish true negatives from inflated zeros). We considered two candidates for the proxy $W$: (i) adjudicator access to the shared electronic health record system EPIC, (ii) either adjudicator access to EPIC, or the statewide health information exchange CRISP. Since $R$ encodes the state of knowing all required information from all sources, we have $R \rightarrow W$.

Our working model is the proxy-augmented ZI MAR under assumptions **A1**\* and **A2**\*, shown in Fig. 2 (b). Using the analytic bounds for the ZI MAR model derived in Section 3.4, we perform a sensitivity analysis to understand how the true CLABSI rate $p(X^{(1)} = 1)$ changes as the proxy-indicator relationship $p(W \mid R)$ varies, within its compatibility range. First, we use the EM algorithm [Dempster et al., 1977] to maximize the observed data likelihood $\hat{\mathcal{L}}_{EM}(X, W, C)$ defined via the full data distribution consistent with our assumptions. Next, we invoke Theorem 5 to obtain the plug-in estimate for $p(W = 0|R = 1)$ and the bounds for $p(W = 0|R = 0)$. Finally, we do a grid search over the bounds interval, compute the full data distribu-

tion $p(R, X, W, C)$ for each value of $p(W = 0|R = 0)$ via (2) and obtain $p(X^{(1)})$ using standard g-formula adjustment in MAR models. The sensitivity analysis curve is shown in Fig 4.

The values of $p(W = 0|R = 0)$ consistent with the model show that inability to make a CLABSI determination is strongly associated with access to patient data via electronic health records. Our obtained (sharp) bound for the nuisance parameter $p(W = 0|R = 0)$ is $[0.89, 1]$, yielding the estimated range of the true CLABSI rate to be $[0.68, 0.78]$. Compared with the baseline rate of 65% under no-zero-inflation assumption, the rate's bound implies that anywhere from 3% to 13% of true CLABSI cases are undercounted, even in our patient cohort with a highly elevated CLABSI prevalence.

We have repeated the analysis using the proxy-augmented ZI MAR model under assumptions **A1**† and **A2**†, shown in Fig. 2 (c). In this case, bounds for $p(W = 0|R = 0, c)$ were obtained, for each value $c$. The narrowest bound $[0.9, 1.0]$ corresponds to patients receiving outpatient parenteral antimicrobial therapy via a peripherally inserted central catheter. On the other hand, the widest bound $[0.67, 1.0]$ corresponds to patients receiving chemotherapy via PICC, a particular type of catheter. We performed a search over the polytope comprised of these bounds and find the estimated range of true CLABSI rate to be $[0.68, 1.0]$. That is, anywhere from 3%-35% of true CLABSI cases are undercounted.

All bounds derived by our methods for the true CLABSI rate were deemed to be medically plausible by our medical collaborators.

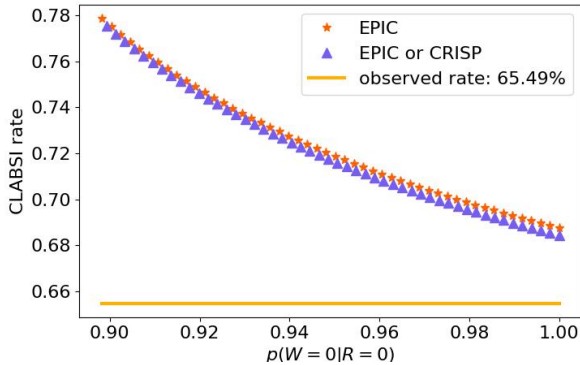

Figure 4: CLABSI rate consistent with model compatible distributions $p(W \mid R)$ under the ZI MAR model with assumptions **A1**\*, and **A2**\*.

# 5 CONCLUSION

In this paper, we considered inference on data with inflated zeros as a missing data problem where censored realizations

are indicated by a 0 rather than by a special token such as "?". This leads to a situation where the censoring indicator for a variable is unobserved any time the value 0 is observed for such a variable. We have shown that this significantly complicates the problem, and results in lack of identification even in simple missing data models such as MCAR.

To address this, we proposed a generalization of the approach in Kuroki and Pearl [2014] which assumes the existence of an informative proxy for the censoring indicator. We show that only some relationships between this proxy and the indicator are compatible with the model, derive analytic bounds for this relationship in a number of cases, and show that in some cases our bound is sharp. Our bounds directly imply bounds on the zero inflated mean parameter. We verified our results by deriving bounds numerically using the `autobounds` package described in Duarte et al. [2023]. Finally, we applied our methods to CLABSI data, which exhibits significant zero inflation. Our methods led to informative bounds on the true CLABSI rate, and provided a natural sensitivity analysis strategy.

Zero inflation is common in many types of data, particularly in electronic health records. Our approach provides a principled strategy for deriving informative conclusions from such data without reliance on unrealistic modeling assumptions.

## Acknowledgements

This research is funded in part by ONR N00014-21-1-2820, NSF 2040804, NSF CAREER 1942239, NIH R01 AI127271-01A1, AHRQ R01 HS027819.

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

# Supplementary Material

**Trung Phung**[*1] **Jaron J.R. Lee**[1] **Opeyemi Oladapo-Shittu**[2] **Eili Y. Klein**[2] **Ayse Pinar Gurses**[1,2,3,4] **Susan M. Hannum**[3] **Kimberly Weems**[5,6] **Jill A. Marsteller**[3,4] **Sara E. Cosgrove**[2,4,5] **Sara C. Keller**[2,4] **Ilya Shpitser**[1]

[1]Johns Hopkins Whiting School of Engineering, Baltimore, MD
[2]Johns Hopkins University School of Medicine, Baltimore, MD
[3]Johns Hopkins Bloomberg School of Public Health, Baltimore, MD
[4]Johns Hopkins Medicine, Baltimore, MD
[5]Johns Hopkins Health System, Baltimore, MD
[6]Vassar Brothers Medical Center, Poughkeepsie, NY

## A  PROOFS

### A.1  DOWNSTREAM IDENTIFICATION

**Proposition 1.** *The full law $p\left(X^{(1)}, R, W, C, X\right)$ exhibiting zero inflation that is Markov relative to an m-DAG $\mathcal{G}$ is identified given $p(R, X, W, C)$ if and only if $\mathcal{G}$ does not contain edges of the form $X_i^{(1)} \rightarrow R_i$ (no self-censoring) and structures of the form $X_j^{(1)} \rightarrow R_i \leftarrow R_j$ (no colluders), and the positivity assumption holds. Moreover, the identifying functional for the full data law coincides with the functional given in Malinsky et al. [2021].*

*Proof.* Following the proof in Nabi et al. [2020], the full law factorizes as

$$
\begin{aligned}
p&\left(R \mid X^{(1)}, C, W\right) \\
&= \frac{1}{Z} \times \prod_{k=1}^{K} p\left(R_k \mid R_{-k} = 1, X^{(1)}, C, W\right) \\
&\quad \times \prod_{R_k, R_l \in R} \mathrm{OR}\left(R_k, R_l \mid R_{-(k,l)} = 1, X^{(1)}, C, W\right) \\
&\quad \times \prod_{R_k, R_l, R_m \in R} f\left(R_k, R_l, R_m \mid R_{-(k,l,m)} = 1, X^{(1)}, C, W\right) \\
&\quad \times \prod_{R_k, R_l, R_m, R_n \in R} f\left(R_k, R_l, R_m, R_n \mid R_{-(k,l,m,n)} = 1, X^{(1)}, C, W\right) \times \cdots \times f\left(R_1, \ldots, R_K \mid X^{(1)}, C, W\right),
\end{aligned}
\tag{12}
$$

where $R_{-k} = R \setminus \{R_k\}$, and similarly for $X^{(1)}$.

- No-collider condition implies $R_k \perp\!\!\!\perp X_k^{(1)} \mid R_{-k}, X_{-k}^{(1)}, C, W$, so $p\left(R_k \mid R_{-k} = 1, X^{(1)}, C, W\right) = p\left(R_k \mid R_{-k} = 1, X_{-k}^{(1)}, C, W\right)$. Hence these factors use only $R = 1$ case of consistency.

- The 2-way odd-ratio $\mathrm{OR}\left(R_k, R_l \mid R_{-(k,l)} = 1, X^{(1)}, C, W\right)$ is not a function of $\{X_k^{(1)}, X_l^{(1)}\}$. Therefore, only case $R = 1$ of consistency is used.

- the 3-way interaction term $f\left(R_k, R_l, R_m, R_n \mid R_{-(k,l,m,n)} = 1, X^{(1)}, C, W\right)$ is not a function of $\{X_k^{(1)}, X_l^{(1)}, X_m^{(1)}\}$. Therefore, only case $R = 1$ of consistency is used. Similarly for any k-way interaction term.

---

[*]tphung1@jhu.edu

Hence the proof in Nabi et al. [2020] applies to ZI problems, whose consistency differs missing data consistency only at $R = 0$ case. □

## A.2 NON-IDENTIFIABILITY PROOF

**Lemma 1.** *Given a ZI model associated with any m-DAG $\mathscr{G}$, both the target law $p(X^{(1)})$ and the full law $p(X^{(1)}, R, C)$ are non-parametrically non-identified.*

*Proof.* Let $\mathscr{G}$ be an m-DAG over $X^{(1)}, R, X, C$ and $\mathscr{P}$ its associated ZI-model. The m-DAG $\mathscr{G}_{\mathrm{mcar}}$ obtained from $\mathscr{G}$ by deleting all edges while keeping $X^{(1)} \to X \leftarrow R$ defines a sub-model $\mathscr{P}_{\mathrm{mcar}} \subseteq \mathscr{P}$ in which $X^{(1)}, R, C$ are jointly independent. If $p(X^{(1)})$ and $p(X^{(1)}, R, C)$ are non-parametrically non-identified in this sub-model, they are also non-identified in $\mathscr{P}$.

It suffices to prove non-identification for binary variables. The target is $p(X^{(1)} = 1)$, and the observed marginals are

$$
\begin{aligned}
p(X = 1)p(c) &= p(X^{(1)} = 1)p(R = 1)p(c) \\
p(X = 0)p(c) &= p(X^{(1)} = 0)p(R = 1)p(c) + p(R = 0)p(c),
\end{aligned}
\tag{13}
$$

using d-separation in $\mathscr{G}_{\mathrm{mcar}}$. Since the second equation is just $p(c)$ minus the first, if the quantity

$$
p(X = 1) = p(X^{(1)} = 1)p(R = 1)
\tag{14}
$$

is shown to be identical for 2 joint distributions in $\mathscr{P}_{\mathrm{mcar}}$, the proof is finished. Indeed, for any $p_1 \in \mathscr{P}_{\mathrm{mcar}}$, we pick any real number $1 > m \geq \max\{p_1(X^{(1)} = 1), p_1(R = 1)\}$ and construct $p_2 \in \mathscr{P}_{\mathrm{mcar}}$ as follow

$$
p_2(X^{(1)} = 1) = \frac{1}{m}p_1(X^{(1)} = 1); \quad p_2(R = 1) = mp_1(R = 1); \quad p_2(C) = p_1(C).
\tag{15}
$$

Evidently, the target laws are different $p_1(X^{(1)}) \neq p_2(X^{(1)})$, yet the observed marginals are the same $p_1(X, C) = p_2(X, C)$. Moreover, the full laws are also different

$$
\begin{aligned}
p_2(X^{(1)} = 0)p_2(R = 1) &= \left(1 - \frac{1}{m}p_1(X^{(1)} = 1)\right)mp_1(R = 1) \\
&\neq p_1(R = 1) - p_1(X^{(1)} = 1)p_1(R = 1) \\
&= p_1(X^{(1)} = 0)p_1(R = 1).
\end{aligned}
\tag{16}
$$

Hence, $p(X^{(1)})$ and $p(X^{(1)}, R, C)$ are non-parametrically non-identified in $\mathscr{P}_{\mathrm{mcar}}$. □

## A.3   EXAMPLES OF COMPATIBILITY ISSUE

Consider the proxy-augmented ZI MCAR model, in which a joint distribution factorizes as

$$p(X^{(1)}, R, X, W) = p(X^{(1)}, R, X)p(W \mid R). \tag{17}$$

Here, the proxy assumptions insist that $p(W = 0 \mid R = 0) \neq p(W = 0 \mid R = 1)$. Therefore, any $p(W \mid R)$ obeys this inequality is said to be model compatible. Moreover, any joint distribution with $p(W \mid R)$ violating this inequality is outside of the model. Works investigate marginal models of hidden variable models often consider this type of compatibility.

In our paper, we mentioned another type of compatibility. Any joint distribution in the model yields a pair of observed law and proxy-indicator conditional distribution $(p(X, W), p(W \mid R))$. Obviously, both $p(X, W)$ and $p(W \mid R)$ produced this way are model compatible. Furthermore, they are compatible to one another, in the sense that there exists a model compatible joint distribution producing them. It is possible to construct an incompatible pair $(p(X, W), p(W \mid R))$ whose components are both model compatible, because the joint distribution yielding them is not in the model. This is illustrated in the following simple examples.

**Example 1:**

$$
\begin{array}{c|c|c}
 & X = 0 & X = 1 \\
\hline
W = 0 & a & b \\
W = 1 & c & d \\
\end{array}
\qquad
\begin{array}{c|c|c}
 & R = 0 & R = 1 \\
\hline
W = 0 & 1 & 0 \\
W = 1 & 0 & 1 \\
\end{array}
\tag{18}
$$

$$\underbrace{\phantom{XXXXXXXXX}}_{p(W,X)} \qquad \underbrace{\phantom{XXXXXXXXX}}_{p(W|R)}$$

Since ZI MCAR does not impose any restriction on $p(W, X)$ in the binary case (see our proof for the bound in the ZI MCAR case), we can pick any number for $a, b, c, d$. In particular, let them be all non-zero. Then both $p(W, X)$ and $p(W \mid R)$ are model compatible. However, there isn't any valid $p(R, X)$ (non-negative, summed to 1) such that the Kuroki-Pearl equation holds $\mathbf{p}_{WX} = \mathbf{p}_{W|R}\mathbf{p}_{RX}$. Attempting to invert $\mathbf{p}_{W|R}$ in this equation will yield negative-valued $p(R, X)$.

**Example 2:**

We choose a joint distribution (DGP) $p(X^{(1)}, R, X, W)$ Markov to the proxy-augmented ZI MCAR graph in Figure 2(a), from which we obtain the true $p(W, X)$, true $p(W \mid R)$, true $p(R, X)$.

We calculate $\hat{p}_1(R, X)$ via the matrix inversion equation using the true $p(W, X)$ and the true $p(W \mid R)$. The calculated $\hat{p}_1(R, X)$ is valid, and close to the true $p(R, X)$ up to floating point precision. This indicates the true $p(W, X)$ and the true $p(W \mid R)$ are compatible to one another.

We sample 100000 data points $(W_i, X_i)$ from this DGP and estimate $\hat{p}(W, X)$ by counting, which is the MLE for binary data. Again, this estimation is in the model, since marginal model for $p(W, X)$ is saturated in the binary case. Then, we calculate $\hat{p}_2(R, X)$ via the matrix inversion equation, using the estimated $\hat{p}(W, X)$ and the true $p(W \mid R)$. This estimated $\hat{p}_2(R, X)$ has a negative value, which renders it invalid.

The code for this experiment could be found in the supplement of the paper. Its output is printed below.

```
True p(W,X):
 [[0.42643891  0.31215362]
  [0.14620603  0.11520144]]
True p(W|R):
 [[0.74919143  0.73043156]
  [0.25080857  0.26956844]]
True p(R,X):
 [[0.43502295  0.          ]
  [0.13762199  0.42735506]]
Computed p(R,X) via matrix inv using true p(W,X) and true p(W|R):
 [[0.43502294 -1.81411279e-16]
  [0.13762199  0.42735505]]
```

```
Estimated p(W,X):
 [[0.42883      0.30976]
  [0.14496      0.11645]]
Computed p(R,X) via matrix inv using estimated p(W,X) and true p(W|R):
 [[ 0.5178968 -0.08300896]
  [ 0.05589317  0.50921896]]
```

## A.4 ZI MCAR MODEL AND BOUNDS

In this section, $X^{(1)}$ and $C$ are categorical, while $R$ and $W$ are binary.

### A.4.1 Model definition

Both the ZI MCAR model and ZI MAR model are Cartesian products, between $p(W \mid R)$ model and $p(X^{(1)}, R, X)$ model, or $p(X^{(1)}, R, X, C)$ model, respectively. Firstly, the adjustment formula establishes a 1-to-1 relation between the $p(X^{(1)}, R, X, C)$ model and the $p(R, X, C)$ model. The constraint of the latter is fully understood.

**Lemma 4.** *C For 1 variable ZI MCAR and ZI MAR model, the full law model for $p(X^{(1)}, R, X, C)$ is 1-to-1 to the model for $p(R, X, C)$ satisfying* **Z**: $\forall x \neq 0, \forall c, p(X = x, R = 0, C = c) = 0$.

*Proof.* We only need to prove the lemma for ZI MAR model.

- $\mathscr{P}$ includes all full laws $p(X^{(1)}, R, X, C)$ factorizing as

$$p(X^{(1)}, R, X, C) = p(X \mid X^{(1)}, R) p(X^{(1)} \mid C) p(R \mid C) p(C), \tag{19}$$

  with $p(X \mid X^{(1)}, R)$ denotes the deterministic ZI-consistency.
- $\mathscr{Q}$ includes all laws $p(R, X, C)$ factorizing as

$$p(R, X, C) = p(X \mid R, C) p(R \mid C) p(C). \tag{20}$$

  and obeying **Z**: $\forall c, \forall x \neq 0 : p(X = x, R = 0, C = c) = 0$.

These 2 models are 1-to-1:

- $(\mathscr{P} \mapsto \mathscr{Q})$: This is just summation $p(R, X, C) = \sum_{X^{(1)}} p(X^{(1)}, R, X, C)$. The ZI-consistency implies **Z**.
- $(\mathscr{Q} \mapsto \mathscr{P})$: By d-separation $p(X^{(1)} \mid C) = p(X \mid R = 1, C)$.

$\square$

In principle, asking whether $p(W \mid R)$ and $p(X, W, C)$ are compatible means pointing out a full law $p(X^{(1)}, R, X, C) p(W \mid R)$ which yields both of them. The above lemma allows us to reformulate this compatibility question by pointing out a joint $p(R, X, C) p(W \mid R)$ in the model. This has the advantage of simplyfying the original compatibility question, i.e., the polynomial program describing it is of higher degree. Moreover, we do not sacrify bound sharpness as we invoke this lemma, since the joint $p(R, X, C) p(W \mid R)$ satisfying **Z** is 1-to-1 to the full law.

**Lemma 5.** *The ZI MCAR model with categorical X and binary R, W, which is Markov to the proxy-augmented Fig. 2 (a) (reproduced in Fig. 5) under proxy assumptions* **A1**, **A2***, is described by*

$$\mathscr{P} = \left\{ (\mathbf{p}_{W|R}, \mathbf{p}_{RX}) \;\middle|\; \begin{array}{l} \mathbf{p}_{W|R} \geq 0, \forall r \left( \sum_w p_{w|r} = 1 \right), p_{w_0|r_0} \neq p_{w_0|r_1}, \\ \mathbf{p}_{RX} \geq 0, \sum_{rx} p_{rx} = 1, \forall x \neq 0 (p_{r_0 x} = 0) \end{array} \right\}. \tag{21}$$

*Proof.* Due to **A1**,

$$p(X^{(1)}, R, X, W) = p(X^{(1)}, R, X) p(W \mid R). \tag{22}$$

Therefore, model for $p(X^{(1)}, R, X, W)$ is a Cartesian product between the model for $p(X^{(1)}, R, X)$ and the model for $p(W \mid R)$. The former is shown to be 1-to-1 to the model for $p(R, X)$ with restriction **Z**, by lemma 4.

$$\left\{ \mathbf{p}_{R,X} \;\middle|\; \mathbf{p}_{R,X} \geq 0, \sum_{x,r} p_{R,X} = 1, \forall x \neq 0 (p_{r_0,x} = 0) \right\}. \tag{23}$$

While the latter is

$$\left\{ \mathbf{p}_{W|R} \;\middle|\; \mathbf{p}_{W|R} \geq 0, \forall r \left( \sum_w p_{w|r} = 1 \right), \det \mathbf{p}_{W|R} \neq 0 \right\}. \tag{24}$$

We just need to rewrite $\det \mathbf{p}_{W|R} \neq 0$. Since $W, R$ are binary

$$
\begin{aligned}
\det \mathbf{p}_{W|R} &= p_{w_0|r_0} p_{w_1|r_1} - p_{w_0|r_1} p_{w_1|r_0} \\
&= p_{w_0|r_0}(1 - p_{w_0|r_1}) - p_{w_0|r_1}(1 - p_{w_0|r_0}) \\
&= p_{w_0|r_0} - p_{w_0|r_1} \neq 0.
\end{aligned}
\tag{25}
$$

$\square$

### A.4.2 Bounds for ZI MCAR

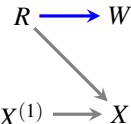

Figure 5: The graph considered in Theorem 3: proxy-augmented ZI MCAR model satisfying **A1** and **A2** (Fig. 2 a in the main paper).

Before proving the bound theorem, we have the following useful lemma:

**Lemma 6.** *For the ZI MCAR model in Theorem 3, $\mathbf{Z}$ constraint $\forall x \neq 0 (q_{r_0 x} = 0)$ is equivalent to $\forall x \neq 0, (q_{w_0|r_1} = q_{w_0|x})$. This means: (i) there is a marginal constraint $\forall x \neq 0 (q_{w_0|x} = q_{w_0|x_1})$, and (ii) $q_{w_0|r_1}$ is point-identified.*

*Proof.* ($\Rightarrow$) direction: Suppose $\forall x \neq 0 (q_{r_0 x} = 0)$. Then for all $x \neq 0$

$$
q_{r_0 x} = 0 \quad \Leftrightarrow \quad q_x = q_{r_0,x} + q_{r_1,x} = q_{r_1,x} \quad \Leftrightarrow \quad q_{r_1|x} = 1.
\tag{26}
$$

Then, for all $x \neq 0$

$$
\begin{aligned}
q_{w_0,x} &= q_{w_0|r_0} q_{r_0,x} + q_{w_0|r_1} q_{r_1,x} = 0 + q_{w_0|r_1} q_{r_1,x} \\
&\Rightarrow q_{w_0|x} = q_{w_0|r_1} q_{r_1|x} = q_{w_0|r_1}.
\end{aligned}
\tag{27}
$$

($\Leftarrow$) direction: Suppose $\forall x \neq 0 (q_{w_0|r_1} = q_{w_0|x})$. Then, for all $x \neq 0$

$$
\begin{aligned}
q_{w_0,x} &= q_{w_0|r_0} q_{r_0,x} + q_{w_0|r_1} q_{r_1,x} \\
&\Rightarrow q_{w_0|x} = q_{w_0|r_0} q_{r_0|x} + q_{w_0|r_1} q_{r_1|x} \\
&\Rightarrow q_{w_0|r_1} = q_{w_0|r_0} q_{r_0|x} + q_{w_0|r_1} q_{r_1|x} \\
&\Rightarrow 0 = (q_{w_0|r_0} - q_{w_0|r_1}) q_{r_0|x}.
\end{aligned}
\tag{28}
$$

Since $q_{w_0|r_0} \neq q_{w_0|r_1}$, we must have $q_{r_0|x} = 0 \Rightarrow q_{r_0,x} = 0$. This is true for all $x \neq 0$.

$\square$

Due to this lemma, an observed law $p(X, W)$ is consistent to the model if and only if $\forall x \neq 0, p(W = 0, X = x) = p(W = 0, X = 1)$. We also require positivity $\forall x, p(X = x) > 0$.

**Theorem 3.** *Consider a ZI MCAR model in Fig. 2 (a) (reproduced in Fig. 5) under proxy assumptions **A1**, **A2**, with categorical $X$ and binary $R, W$. Given a consistent observed law $p(X, W)$ satisfying positivity, the set of compatible proxy-indicator conditionals $q(W \mid R)$ is given by*

$$
q_{w_0|r_1} = p_{w_0|x_1}
$$

$$
q_{w_0|r_0} \in
\begin{cases}
[p_{w_0|x_0}, 1] & \text{if } p_{w_0|x_0} > p_{w_0|x_1} \\
[0, p_{w_0|x_0}] & \text{if } p_{w_0|x_0} < p_{w_0|x_1} \\
(0, 1) \setminus \{p_{w_0|x_0}\} & \text{if } p_{w_0|x_0} = p_{w_0|x_1}
\end{cases}
$$

*These bounds are sharp. Moreover, if $p_{w_0|x_0} = p_{w_0|x_1}$, $p(X, W)$ must satisfy $0 < p_{w_0|x_0} < 1$, and zero inflation does not occur, i.e., $q(R = 0) = 0$.*

*Proof.* In the following, $q(\cdot)$ denotes an element in a model, while $p(\cdot)$ is derived from the given marginal $p(X,W)$.

In principle, any compatible $q(W \mid R)$ to $p(X,W)$ must be derived from some full joint distribution $q(X^{(1)},R,X)q(W \mid R)$, such that $q(X,W) = p(X,W)$. Since the model for $q(X^{(1)},R,X)$ is 1-to-1 to the model for $q(R,X)$ with restriction **Z**, we can simplify this process by considering the marginal model for $q(R,X,W)$

$$\mathscr{P} = \left\{ (\mathbf{q}_{W|R}, \mathbf{q}_{RX}) \;\middle|\; \begin{array}{l} \mathbf{q}_{W|R} \geq 0, \forall r \left( \sum_w q_{w|r} = 1 \right), q_{w_0|r_0} \neq q_{w_0|r_1}, \\ \mathbf{q}_{RX} \geq 0, \sum_{rx} q_{rx} = 1, \forall x \neq 0(q_{r_0 x} = 0) \end{array} \right\}. \tag{29}$$

The subset of $q(R,X,W)$ yielding the observed law $p(X,W)$ is

$$\mathscr{Q} = \left\{ (\mathbf{q}_{W|R}, \mathbf{q}_{RX}) \mid \mathbf{q}_{W|R}\mathbf{q}_{RX} = \mathbf{p}_{WX}, (\mathbf{q}_{W|R}, \mathbf{q}_{RX}) \in \mathscr{P} \right\}. \tag{30}$$

**Polynomial program**

Since $\mathbf{q}_{W|R} \in \mathscr{P}$ is invertible, $\mathscr{Q}$ is the set of all pairs $(\mathbf{q}_{W|R}, \mathbf{q}_{RX})$ with $\mathbf{q}_{RX} = [\mathbf{q}_{W|R}]^{-1}\mathbf{p}_{WX}$ and $\mathbf{q}_{W|R} \in \mathscr{B}$,

$$\mathscr{B} = \left\{ \mathbf{q}_{W|R} \;\middle|\; \begin{array}{l} \mathbf{q}_{W|R} \geq 0, \forall r \left( \sum_w q_{w|r} = 1 \right), q_{w_0|r_0} \neq q_{w_0|r_1}, \\ \mathbf{q}_{RX} \geq 0, \sum_{rx} q_{rx} = 1, \forall x \neq 0(q_{r_0 x} = 0), \\ \text{where } \mathbf{q}_{RX} = [\mathbf{q}_{W|R}]^{-1}\mathbf{p}_{WX}. \end{array} \right\}. \tag{31}$$

$\mathscr{B}$ is called the compatibility set of $q(W \mid R)$ w.r.t. $p(X,W)$. As mentioned in the main paper, one can directly solve for $\mathscr{B}$ via the following polynomial program, where $\mathbf{q}_{RX}$ are slack variables.

$$\begin{aligned} \max_{q_{w_0|r_0}} \quad & \pm q_{w_0|r_0} \\ \text{s.t.} \quad & \mathbf{q}_{W|R}\mathbf{q}_{RX} = \mathbf{p}_{WX}, \\ & \mathbf{q}_{W|R} \geq 0, \forall r(\sum_w q_{w|r} = 1), q_{w_0|r_0} \neq q_{w_0|r_1}, \\ & \mathbf{q}_{RX} \geq 0, \sum_{rx} q_{rx} = 1, \forall x \neq 0(q_{r_0 x} = 0). \end{aligned} \tag{32}$$

As we will show below, the constraint $\forall x \neq 0(q_{r_0 x} = 0)$ is equivalent to $q_{w_0|r_1} = p_{w_0|x_1}$. This is a quadratic program due to the first constraint.

**Linear program**

We will simplify $\mathscr{B}$. We do so by considering its superset and adding constraints to it. Firstly, $\mathscr{B}$ could be parameterized by only 2 numbers, because its superset is

$$\left\{ \mathbf{q}_{W|R} \mid \mathbf{q}_{W|R} \geq 0, \forall r \left( \sum_w q_{w|r} = 1 \right) \right\} = \left\{ \mathbf{q}_{W|R} = \begin{pmatrix} q_{w_0|r_0} & q_{w_0|r_1} \\ 1 - q_{w_0|r_0} & 1 - q_{w_0|r_1} \end{pmatrix} \;\middle|\; 0 \leq q_{w_0|r_0} \leq 1, 0 \leq q_{w_0|r_1} \leq 1 \right\}. \tag{33}$$

Secondly, when $q_{w_0|r_0} \neq q_{w_0|r_1}$, the 2-2 matrix $\mathbf{q}_{W|R}$ has inverse

$$[\mathbf{q}_{W|R}]^{-1} = \frac{1}{q_{w_0|r_0} - q_{w_0|r_1}} \underbrace{\begin{pmatrix} 1 - q_{w_0|r_1} & -q_{w_0|r_1} \\ q_{w_0|r_0} - 1 & q_{w_0|r_0} \end{pmatrix}}_{M}. \tag{34}$$

Therefore, we can transform the quadratic constraint into the following equivalent linear constraints

$$\begin{aligned} & \left\{ \mathbf{q}_{W|R} \mid \mathbf{q}_{W|R} \geq 0, \forall r \left( \sum_w q_{w|r} = 1 \right), q_{w_0|r_0} \neq q_{w_0|r_1}, \mathbf{q}_{RX} = [\mathbf{q}_{W|R}]^{-1}\mathbf{p}_{WX} \geq 0 \right\} \\ & = \bigcup_{s \in \{1,-1\}} \left\{ \mathbf{q}_{W|R} \mid \mathbf{q}_{W|R} \geq 0, \forall r \left( \sum_w q_{w|r} = 1 \right), sq_{w_0|r_0} > sq_{w_0|r_1}, sM\mathbf{p}_{WX} \geq 0 \right\}. \end{aligned} \tag{35}$$

Next, given $\mathbf{q}_{W|R}\mathbf{q}_{RX} = \mathbf{p}_{WX}$ where all terms are non-negative, $\sum_w q_{w|R} = 1$, $\sum_{w,x} p_{w,x} = 1$, and $\mathbf{q}_{W|R}$ is invertible, then $\sum_{rx} q_{rx} = 1$. Hence, $\sum_{rx} q_{rx} = 1$ is a redundant constraint. *Proof:* entry $ij$-th $[\mathbf{p}_{WX}]_{ij} = q_{w_i|r_0}q_{r_0 x_j} + q_{w_i|r_1}q_{r_1 x_j}$. Then $1 = \sum_{ij} [\mathbf{p}_{WX}]_{ij} = \sum_j \left( (\sum_i q_{w_i|r_0})q_{r_0 x_j} + (\sum_i q_{w_i|r_1})q_{r_1 x_j} \right) = \sum_j (q_{r_0 x_j} + q_{r_1 x_j})$.

Finally, lemma 6 says $\forall x \neq 0(q_{r_0 x} = 0) \Leftrightarrow \forall x \neq 0(q_{w_0|r_1} = q_{w_0|x})$, and $q_{w_0|x} = p_{w_0|x}$ in $\mathscr{B}$. Note that this lemma also requires $p(X,W)$ to satisfy the marginal constraint $p_{w_0|x} = p_{w_0|x_1}$. Therefore, the constraint $\forall x \neq 0(q_{r_0 x} = 0) \Leftrightarrow q_{w_0|r_1} = p_{w_0|x_1}$.

Putting together, we can write $\mathscr{B}$ as

$$\mathscr{B} = \bigcup_{s\in\{1,-1\}} \left\{ \mathbf{q}_{W|R} = \begin{pmatrix} q_{w_0|r_0} & q_{w_0|r_1} \\ 1-q_{w_0|r_0} & 1-q_{w_0|r_1} \end{pmatrix} \middle| \begin{array}{l} s\begin{pmatrix} 1-q_{w_0|r_1} & -q_{w_0|r_1} \\ q_{w_0|r_0}-1 & q_{w_0|r_0} \end{pmatrix}\mathbf{p}_{WX} \geq \mathbf{0}, \\ s\cdot q_{w_0|r_0} > s\cdot q_{w_0|r_1}, 0\leq q_{w_0|r_0}\leq 1, \\ q_{w_0|r_1} = p_{w_0|x_1} \end{array} \right\}. \tag{36}$$

or,

$$\mathscr{B} = \left\{ \mathbf{q}_{W|R} = \begin{pmatrix} q_{w_0|r_0} & q_{w_0|r_1} \\ 1-q_{w_0|r_0} & 1-q_{w_0|r_1} \end{pmatrix} \middle| q_{w_0|r_1} = p_{w_0|x_1}, q_{w_0|r_0} \in \mathscr{B}_{w_0|r_0} \right\},$$

$$\mathscr{B}_{w_0|r_0} = \bigcup_{s\in\{1,-1\}} \mathscr{B}^s_{w_0|r_0} = \bigcup_{s\in\{1,-1\}} \left\{ q_{w_0|r_0} \middle| \begin{array}{l} s\begin{pmatrix} 1-q_{w_0|r_1} & -q_{w_0|r_1} \\ q_{w_0|r_0}-1 & q_{w_0|r_0} \end{pmatrix}\mathbf{p}_{WX} \geq \mathbf{0}, \\ s\cdot q_{w_0|r_0} > s\cdot q_{w_0|r_1}, 0\leq q_{w_0|r_0}\leq 1, \\ q_{w_0|r_1} = p_{w_0|x_1} \end{array} \right\}. \tag{37}$$

The set $\mathscr{B}_{w_0|r_0}$ is called the **compatible set** of $q_{w_0|r_0}$ w.r.t. $p(X,W)$. As will be shown, this is an interval in $[0,1]$, hence the name **compatibility bound**.

To find $\mathscr{B}_{w_0|r_0}$, we will find each $\mathscr{B}^s_{w_0|r_0}$ and take their union. Each $\mathscr{B}^s_{w_0|r_0}$ could be numerically computed by solving the 2 linear programs

$$\begin{aligned} \max_{q_{w_0|r_0}} \quad & \pm q_{w_0|r_0} \\ \text{s.t.} \quad & s\cdot\begin{pmatrix} 1-q_{w_0|r_1} & -q_{w_0|r_1} \\ q_{w_0|r_0}-1 & q_{w_0|r_0} \end{pmatrix}\mathbf{p}_{WX} \geq \mathbf{0}, \\ & s\cdot q_{w_0|r_0} > s\cdot q_{w_0|r_1}, 0\leq q_{w_0|r_0}\leq 1, \\ & q_{w_0|r_1} = p_{w_0|x_1}. \end{aligned} \tag{38}$$

These problems are linear program as $q_{w_0|r_0}$ is the only unknown and all constraints are linear. The set $\mathscr{B}^s_{w_0|r_0}$ is the interval whose endpoints are 2 numbers returned by these programs.

**Solutions to linear programs**

*Solving $\mathscr{B}^{s=1}_{w_0|r_0}$:* We expand the matrix multiplication equation

$$\begin{aligned} & p_{w_0,x_0}\left(1-q_{w_0|r_1}\right)-p_{w_1,x_0}q_{w_0|r_1}\geq 0 & \Leftrightarrow & \quad p_{w_0|x_0}p_{w_1|x_1}-p_{w_1|x_0}p_{w_0|x_1}\geq 0 \\ \forall x\neq 0, \quad & p_{w_0,x}\left(1-q_{w_0|r_1}\right)-p_{w_1,x}q_{w_0|r_1}\geq 0 & \Leftrightarrow & \quad \forall x\neq 0, p_{w_0,x}p_{w_1|x_1}-p_{w_1,x}p_{w_0|x_1}=0 \\ & p_{w_0,x_0}\left(q_{w_0|r_0}-1\right)+p_{w_1,x_0}q_{w_0|r_0}\geq 0 & \Leftrightarrow & \quad q_{w_0|r_0}\geq \frac{p_{w_0,x_0}}{p_{w_0,x_0}+p_{w_1,x_0}}=p_{w_0|x_0} \\ \forall x\neq 0, \quad & p_{w_0,x}\left(q_{w_0|r_0}-1\right)+p_{w_1,x}q_{w_0|r_0}\geq 0 & \Leftrightarrow & \quad \forall x\neq 0, q_{w_0|r_0}\geq \frac{p_{w_0,x}}{p_{w_0,x}+p_{w_1,x}}=p_{w_0|x}. \end{aligned} \tag{39}$$

In the derivations above, we use the positivity assumption $\forall x(p(X=x)>0)$. At the very least, we assume there is zeros, i.e., $p(X=0)>0$, otherwise the problem does not make sense. If positivity is violated, e.g., $\exists x\neq 0, p(X=x)=0$, one can show that $p_{w_0,x}=p_{w_1,x}=0$, and hence this value $x$ does not place any restriction on $q_{w_0|r_0}$, and can be ignored in the following discussion.

The first equation shows that the $s=1$ case has no solution if $p_{w_0|x_0}p_{w_1|x_1}-p_{w_1|x_0}p_{w_0|x_1}<0$. When the LHS is non-negative, the feasible region $\mathscr{B}^{s=1}_{w_0|r_0}$ is $\max_x p_{w_0|x}\leq q_{w_0|r_0}\leq 1$. We can further split into 2 cases, and note that $q_{w_0|r_0}>q_{w_0|r_1}$, per $s=1$.

1. If $p_{w_0|x_0}p_{w_1|x_1}-p_{w_1|x_0}p_{w_0|x_1}=0 \Leftrightarrow p_{w_0|x_0}=p_{w_0|x_1}$, which is true for all values in $[0,1]$. Then $p_{w_0|x_0}<q_{w_0|r_0}\leq 1$. For this to make sense, we must have $p_{w_0|x_0}<1$.

2. If $p_{w_0|x_0}p_{w_1|x_1}-p_{w_1|x_0}p_{w_0|x_1}>0 \Leftrightarrow p_{w_0|x_0}>p_{w_0|x_1}$, which is true for all $0\leq p_{w_0|x_1}<1$. Then $p_{w_0|x_0}\leq q_{w_0|r_0}\leq 1$.

The bounds are sharp because they are the feasible regions $\mathscr{B}^{s=1}_{w_0|r_0}$.

*Solving $\mathscr{B}^{s=-1}_{w_0|r_0}$*: Similarly

$$
\begin{aligned}
p_{w_0,x_0}\left(1-q_{w_0|r_1}\right)-p_{w_1,x_0}q_{w_0|r_1} \le 0 &\Leftrightarrow p_{w_0|x_0}p_{w_1|x_1}-p_{w_1|x_0}p_{w_0|x_1} \le 0 \\
\forall x \ne 0, \quad p_{w_0,x}\left(1-q_{w_0|r_1}\right)-p_{w_1,x}q_{w_0|r_1} \le 0 &\Leftrightarrow \forall x \ne 0, p_{w_0,x}p_{w_1|x_1}-p_{w_1,x}p_{w_0|x_1} = 0 \\
p_{w_0,x_0}\left(q_{w_0|r_0}-1\right)+p_{w_1,x_0}q_{w_0|r_0} \le 0 &\Leftrightarrow q_{w_0|r_0} \le \frac{p_{w_0,x_0}}{p_{w_0,x_0}+p_{w_1,x_0}} = p_{w_0|x_0} \\
\forall x \ne 0, \quad p_{w_0,x}\left(q_{w_0|r_0}-1\right)+p_{w_1,x}q_{w_0|r_0} \le 0 &\Leftrightarrow \forall x \ne 0, q_{w_0|r_0} \le \frac{p_{w_0,x}}{p_{w_0,x}+p_{w_1,x}} = p_{w_0|x}.
\end{aligned}
\tag{40}
$$

The first equation shows that the $s=1$ case has no solution if $p_{w_0|x_0}p_{w_1|x_1}-p_{w_1|x_0}p_{w_0|x_1} > 0$. When the LHS is non-positive, the feasible region $\mathscr{B}^{s=-1}_{w_0|r_0}$ is $0 \le q_{w_0|r_0} \le \min_x p_{w_0|x}$. We can further split into 2 cases, and note that $q_{w_0|r_0} < q_{w_0|r_1}$, per $s=-1$.

1. If $p_{w_0|x_0}p_{w_1|x_1}-p_{w_1|x_0}p_{w_0|x_1} = 0 \Leftrightarrow p_{w_0|x_0} = p_{w_0|x_1}$, which is true for all values in $[0,1]$. Then $0 \le q_{w_0|r_0} < p_{w_0|x_0}$. For this to make sense, we must have $p_{w_0|x_0} > 0$.

2. If $p_{w_0|x_0}p_{w_1|x_1}-p_{w_1|x_0}p_{w_0|x_1} < 0 \Leftrightarrow p_{w_0|x_0} < p_{w_0|x_1}$. which is true for all $0 < p_{w_0|x_1} \le 1$. Then $0 \le q_{w_0|r_0} \le p_{w_0|x_0}$.

The bounds are sharp because they are the feasible regions $\mathscr{B}^{s=-1}_{w_0|r_0}$.

**Result**

Combine these results to get the compatibility bound $\mathscr{B}_{w_0|r_0} = \bigcup_{s\in\{1,-1\}} \mathscr{B}^{s}_{w_0|r_0}$. The bounds are sharp.

$$
q_{w_0|r_1} = p_{w_0|x_1}
$$

$$
q_{w_0|r_0} \in \begin{cases} [p_{w_0|x_0}, 1] \text{ if } p_{w_0|x_0} > p_{w_0|x_1} \\ [0, p_{w_0|x_0}] \text{ if } p_{w_0|x_0} < p_{w_0|x_1} \\ (0,1) \setminus \{p_{w_0|x_0}\} \text{ if } 0 < p_{w_0|x_0} = p_{w_0|x_1} < 1 \end{cases}
$$

The situations $p_{w_0|x_0} = p_{w_0|x_1} \in \{0,1\}$ are not allowed by the model. Moreover, if $p_{w_0|x_0} = p_{w_0|x_1}$ then $q(R=0)=0$, i.e., zero inflation does not occur. *Proof*:

$$
\begin{aligned}
p_{w_0|x_0} &= q_{w_0|r_0}q_{r_0|x_0}+q_{w_0|r_1}q_{r_1|x_0} \\
p_{w_0|x_1} &= q_{w_0|r_1} \quad \text{(id)}
\end{aligned}
\tag{41}
$$

Therefore, subtracting both sides,

$$
0 = p_{w_0|x_0}-p_{w_0|x_1} = q_{w_0|r_0}q_{r_0|x_0}-q_{w_0|r_1}q_{r_0|x_0} = \left(q_{w_0|r_0}-q_{w_0|r_1}\right)q_{r_0|x_0}.
\tag{42}
$$

Due to proxy assumption **A2**: $q_{w_0|r_0} \ne q_{w_0|r_1}$. Then the LHS equals 0 if and only if $q_{r_0|x_0} = 0$. Moreover, **Z** implies $\forall x \ne 0, q_{r_0|x} = 0$. Then $q_{r_0} = \sum_x q_{r_0|x}p_x = 0$.

$\square$

## A.5 ZI MAR PROOFS

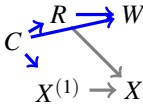

Figure 6: The graph considered in Theorem 4: proxy-augmented ZI MAR model satisfying $\mathbf{A1}^{\dagger}$ and $\mathbf{A2}^{\dagger}$ (Fig. 2 (b) in the main paper).

**Theorem 4.** *Consider a ZI MAR model in Fig. 2 (b) (reproduced in Fig. 6) under proxy assumptions $\mathbf{A1}^{\dagger}$ and $\mathbf{A2}^{\dagger}$, with categorical $X, C$ and binary $R, W$. Given a consistent observed law $p(X, W, C)$ satisfying positivity, the set of compatible proxy-indicator conditional distributions $q(W \mid R, C)$ is given by, for each value $c$,*

$$q_{w_0|r_1,c} = p_{w_0|x_1,c}$$

$$q_{w_0|r_0,c} \in \begin{cases} [p_{w_0|x_0,c}, 1] & \text{if } p_{w_0|x_0,c} > p_{w_0|x_1,c} \\ [0, p_{w_0|x_0,c}] & \text{if } p_{w_0|x_0,c} < p_{w_0|x_1,c} \\ (0,1) \setminus \{p_{w_0,|x_0,c}\} & \text{if } p_{w_0|x_0,c} = p_{w_0|x_1,c} \end{cases}$$

*These bounds are sharp. Moreover, if $p_{w_0|x_0,c} = p_{w_0|x_1,c}$, $p(X, W, C)$ must satisfy $0 < p_{w_0|x_0,c} < 1$, and zero inflation does not occur for stratum $C = c$, i.e., $q(R = 0 \mid c) = 0$.*

*Proof.*

**Model definition.**

We assume $C$ is a cardinal variable, taking values in a finite set $\mathscr{C}$. Any joint distribution in this ZI MAR model is

$$q\left(X^{(1)}, R, X, W, C\right) = q\left(X^{(1)}, R, X, W \mid C\right) p(C) \tag{43}$$

Since the Markov factors are variationally independent, the ZI MAR model is a Cartesian product

$$\begin{aligned} \mathscr{P}_{\text{ZI MAR}}^{(1)} &= \left(\bigotimes_{c \in \mathscr{C}} \mathscr{P}_{\text{ZI MCAR}}^{(1)}(c)\right) \otimes \mathscr{P}_C \\ \mathscr{P}_C &= \{q(C)\}, \\ \mathscr{P}_{\text{ZI MCAR}}^{(1)}(c) &= \left\{q(X^{(1)}, R, X, W \mid c) \mid \mathbf{A1}, \mathbf{A2}\right\} \end{aligned} \tag{44}$$

Note how constraints $\mathbf{A1}^{\dagger}$, $\mathbf{A2}^{\dagger}$ are equivalent to imposing $\mathbf{A1}$, $\mathbf{A2}$ to each stratum $C = c$. Notation: (i) $\mathscr{P}_C = \{q(C)\}$ means $\mathscr{P}_C$ is a non-parametric model contains all probability distribution $q(C)$, and (ii) probability constraints are assumed to hold.

In this product, $\mathscr{P}_{\text{ZI MCAR}}^{(1)}(c)$ for all $c$ are the same ZI MCAR model described in Theorem 3, repeated $|\mathscr{C}|$ times. The value $c$ is not a parameter of the model $\mathscr{P}_{\text{ZI MCAR}}^{(1)}(c)$, but a constant. Its only purpose is for the sake of book-keeping when constructing the joint distribution in $\mathscr{P}_{\text{ZI MAR}}^{(1)}$. For MAR, standard adjustment method point identifies $q(X^{(1)}, R, X, W)$ as a functional of $q(R, X, W)$. Therefore, as shown in lemma 4 the set $\mathscr{P}_{\text{ZI MCAR}}^{(1)}(c)$ is 1-to-1 to the set $\mathscr{P}_{\text{ZI MCAR}}(c) = \{q(R, X, W \mid c) \mid \mathbf{Z}, \mathbf{A1}, \mathbf{A2}\}$. Hence, we are interested in the marginal model

$$\mathscr{P}_{\text{ZI MAR}} = (\bigotimes_{c \in \mathscr{C}} \mathscr{P}_{\text{ZI MCAR}}(c)) \otimes \mathscr{P}_C. \tag{45}$$

**Finding compatible set.**

Given an observed law $\mathbf{p}_{WXC}$, we want to find the compatible set w.r.t. this law

$$\mathscr{Q} = \left\{\mathbf{q}_{XRWC} \mid \mathbf{q}_{XRWC} \in \mathscr{P}_{\text{ZI MAR}}, \forall c \in \mathscr{C} \left(\mathbf{q}_{W|Rc} \mathbf{q}_{RX|c} = \mathbf{p}_{WX|c}\right), \forall c \in \mathscr{C} \left(q(c) = p(c)\right)\right\}. \tag{46}$$

Geometrically speaking, this set is the intersection of our model $\mathscr{P}_{\text{ZI MAR}}$ with the constraint set $\mathscr{E}$, which is itself a Cartesian product,

$$\mathscr{E} = \left\{ \mathbf{q}_{XRWC} \mid \forall c \in \mathscr{C} \left( \mathbf{q}_{W|Rc} \mathbf{q}_{RX|c} = \mathbf{p}_{WX|c} \right), \text{ and } \forall c \in \mathscr{C} \left( q(c) = p(c) \right) \right\}$$
$$= \bigotimes_{c \in \mathscr{C}} \left\{ \mathbf{q}_{XRW|c} \mid \mathbf{q}_{W|Rc} \mathbf{q}_{RX|c} = \mathbf{p}_{WX|c} \right\} \otimes \left\{ p(c) \right\}. \tag{47}$$

Here we abuse notation $\{p(c)\}$ to mean the set with 1 element - the observed law $p(C)$, which is not the model $\mathscr{P}_C$.

Since the constraint $\mathbf{q}_{W|Rc} \mathbf{q}_{RX|c} = \mathbf{p}_{WX|c}$ only concerns $q(R,X,W \mid c)$ and does not concern other $q(R,X,W \mid c')$ in any way, we push each constraint to the corresponding $\mathscr{P}_{\text{ZI MCAR}}(c)$. In other words, we will proceed to find the ZI MCAR compatibility bound for each level $c$, as shown below. Mathematically, as Cartesian product could be written as intersection: If $A, C \subseteq U$ and $B, D \subseteq V$, then

$$(A \otimes B) \cap (C \otimes D) = (A \otimes V) \cap (U \otimes B) \cap (C \otimes V) \cap (U \otimes D)$$
$$= (A \cap C \otimes V) \cap (U \otimes (B \cap D)) \tag{48}$$
$$= (A \cap C) \otimes (B \cap D).$$

We could transform

$$\mathscr{Q} = \mathscr{P}_{\text{ZI MAR}} \cap \mathscr{E} = \left( \left( \bigotimes_{c \in \mathscr{C}} \mathscr{P}_{\text{ZI MCAR}}(c) \right) \otimes \mathscr{P}_C \right) \cap \left( \bigotimes_{c \in \mathscr{C}} \left\{ \mathbf{q}_{XRW|c} \mid \mathbf{q}_{W|Rc} \mathbf{q}_{RX|c} = \mathbf{p}_{WX|c} \right\} \otimes \{p(c)\} \right)$$
$$= \left( \left( \bigotimes_{c \in \mathscr{C}} \mathscr{P}_{\text{ZI MCAR}}(c) \right) \cap \left( \bigotimes_{c \in \mathscr{C}} \left\{ \mathbf{q}_{XRW|c} \mid \mathbf{q}_{W|Rc} \mathbf{q}_{RX|c} = \mathbf{p}_{WX|c} \right\} \right) \right) \otimes \{p(c)\} \tag{49}$$
$$= \left( \bigotimes_{c \in \mathscr{C}} \left( \mathscr{P}_{\text{ZI MCAR}}(c) \cap \left\{ \mathbf{q}_{XRW|c} \mid \mathbf{q}_{W|Rc} \mathbf{q}_{RX|c} = \mathbf{p}_{WX|c} \right\} \right) \right) \otimes \{p(c)\}$$
$$= \left( \bigotimes_{c \in \mathscr{C}} \mathscr{Q}_{\text{ZI MCAR}}(c) \right) \otimes \{p(c)\},$$

where

$$\mathscr{Q}_{\text{ZI MCAR}}(c) = \left\{ q(R,X,W \mid c) \mid \mathbf{Z}, \mathbf{A1}, \mathbf{A2}, \mathbf{q}_{W|Rc} \mathbf{q}_{RX|c} = \mathbf{p}_{WX|c} \right\}. \tag{50}$$

This is exactly the set $\mathscr{Q}$ described in Theorem 3. Therefore, this equation suggests the application of Theorem 3 to each stratum $C = c$. First, there are marginal constraints: $\forall c, \forall x \neq 0, p_{w_0|x,c} = p_{w_0|x_1,c}$. Second,

$$\mathscr{Q}_{\text{ZI MCAR}}(c) = \left\{ (\mathbf{q}_{W|R,c}, \mathbf{q}_{RX|c}) \mid \mathbf{q}_{RX|c} = [\mathbf{q}_{W|R|c}]^{-1} \mathbf{p}_{WX|c}, \mathbf{q}_{W|R} \in \mathscr{B}, \right\} \tag{51}$$

where $\mathscr{B}$ contains stochastic matrix $\mathbf{q}_{W|Rc}$ satisfying

$$q_{w_0|r_1,c} = p_{w_0|x_1,c}$$

$$q_{w_0|r_0,c} \in \begin{cases} [p_{w_0|x_0,c}, 1] & \text{if } p_{w_0|x_0,c} > p_{w_0|x_1,c} \\ [0, p_{w_0|x_0,c}] & \text{if } p_{w_0|x_0,c} < p_{w_0|x_1,c} \\ (0,1) \setminus \{p_{w_0|x_0,c}\} & \text{if } 0 < p_{w_0|x_0,c} = p_{w_0|x_1,c} < 1. \end{cases}$$

Moreover, if $p_{w_0|x_0,c} = p_{w_0|x_1,c}$ then $0 < p_{w_0|x_0,c} < 1$ is an additional condition, and zero inflation does not occur for stratum $C = c$, i.e., $q(R = 0 \mid c) = 0$. Since the compatibility set $\mathscr{Q}$ in this case is a Cartesian product of compatibility sets described in Theorem 3, which is sharp, $\mathscr{Q}$ is sharp. $\qed$

Figure 7: The graph considered in Theorem 5: proxy-augmented ZI MAR model satisfying **A1**$^*$ and **A2**$^*$ (Fig. 2 (c) in the main paper).

**Theorem 5.** *Consider a ZI MAR model in Fig. 2 (c) (reproduced in Fig. 7) under proxy assumptions* **A1**$^*$ *and* **A2**$^*$, *with categorical $X,C$ and binary $R,W$. Given a consistent observed law $p(X,W,C)$ satisfying positivity, the set of compatible proxy-indicator conditional distributions $q(W \mid R)$ is given by*

$$q_{w_0|r_1} = p_{w_0|x_1}$$

$$q_{w_0|r_0} \in \begin{cases} [\max_c p_{w_0|x_0,c}, 1] \text{ if } \exists \tilde{c}, p_{w_0|x_0,\tilde{c}} > p_{w_0|x_1} \\ [0, \min_c p_{w_0|x_0,c}] \text{ if } \exists \tilde{c}, p_{w_0|x_0,\tilde{c}} < p_{w_0|x_1} \\ (0,1) \setminus \{p_{w_0|x_1}\} \text{ if } \forall c, p_{w_0|x_0,c} = p_{w_0|x_1} \end{cases}$$

*These bounds are sharp. Moreover, if $\forall c, p_{w_0|x_0,c} = p_{w_0|x_1}$, $p(X,W,C)$ must satisfy $\forall c, 0 < p_{w_0|x_0,c} < 1$, and zero inflation does not occur, i.e., $q(R=0)=0$.*

*Proof.*

**Model definition.** We assume $C$ is a cardinal variable, taking values in a finite set $\mathscr{C}$. Any joint distribution in this ZI MAR model is

$$q\left(X^{(1)},R,X,W,C\right) = q\left(X^{(1)},R,X \mid C\right)q(W \mid R)q(C) \tag{52}$$

Since the Markov factors are variationally independent, the ZI MAR model is a Cartesian product

$$\mathscr{P}^{(1)}_{\text{ZI MAR}} = \left(\otimes_{c\in\mathscr{C}} \mathscr{P}_{X^{(1)}XR}(c)\right) \otimes \mathscr{P}_{W|R} \otimes \mathscr{P}_C$$
$$\mathscr{P}_C = \{q(C)\},$$
$$\mathscr{P}_{W|R} = \{q(W \mid R) \mid \det \mathbf{q}_{W|R} \neq 0\}, \tag{53}$$
$$\mathscr{P}_{X^{(1)}XR}(c) = \left\{q(X^{(1)},R,X \mid c)\right\}$$

Lemma 4 says $\mathscr{P}_{X^{(1)}XR}(c)$ is 1-to-1 to the set $\mathscr{P}_{XR}(c) = \{p(R,X \mid c) \mid \mathbf{Z}\}$. Hence, we are interested in the marginal model

$$\mathscr{P}_{\text{ZI MAR}} = \left(\otimes_{c\in\mathscr{C}} \mathscr{P}_{XR}(c)\right) \otimes \mathscr{P}_{W|R} \otimes \mathscr{P}_C. \tag{54}$$

**Finding compatible set.**

Given observed law $\mathbf{p}_{WXC}$, we want to find the compatible set w.r.t. this law

$$\mathscr{Q} = \left\{\mathbf{q}_{XRWC} \mid \mathbf{q}_{XRWC} \in \mathscr{P}_{\text{ZI MAR}}, \forall c \in \mathscr{C}\left(\mathbf{q}_{W|R}\mathbf{q}_{RX|c} = \mathbf{p}_{WX|c}\right), \forall c \in \mathscr{C}\left(q(c) = p(c)\right)\right\}. \tag{55}$$

This is similar to the compatible set we consider when **A1**$^\dagger$, **A2**$^\dagger$ hold (e.g., when $C \to W$), except the same $\mathbf{q}_{W|R}$ is shared between the constraints $\mathbf{q}_{W|R}\mathbf{q}_{RX|c} = \mathbf{p}_{WX|c}$. Each constraint restricts $\mathbf{q}_{W|R}$ in a different way, hence we cannot write $\mathscr{Q}$ as a Cartesian product to separate the constraints as we did before.

To proceed, note that $\mathscr{Q}$ is 1-to-1 to a set containing only $\mathbf{q}_{W|R}$, just as in ZI MCAR proof.

$$\mathscr{Q} = \left\{(\mathbf{q}_{W|R}, (\mathbf{q}_{RX|c})_{c\in\mathscr{C}}) \mid \forall c\left(\mathbf{q}_{RX|c} = [\mathbf{q}_{W|R}]^{-1}\mathbf{p}_{WX|c}\right), \mathbf{q}_{W|R} \in \mathscr{B}\right\} \otimes \{p(C)\}$$

$$\mathscr{B} = \left\{\mathbf{q}_{W|R} \middle| \begin{array}{c} \mathbf{q}_{W|R} \geq 0, \forall r\left(\sum_w q_{w|r} = 1\right), q_{w_0|r_0} \neq q_{w_0|r_1}, \\ \text{for each } c: \mathbf{q}_{RX|c} \geq 0, \sum_{rx} q_{rx|c} = 1, \forall x \neq 0(q_{r_0xc} = 0), \\ \text{where } \mathbf{q}_{RX|c} = [\mathbf{q}_{W|R}]^{-1}\mathbf{p}_{WX|c}. \end{array}\right\}. \tag{56}$$

This set is the intersection $\mathscr{B} = \cap_{c \in \mathscr{C}} \mathscr{B}_c$, in which each $\mathscr{B}_c$ contains only constraints associated with values $c$.

$$\mathscr{B}_c = \left\{ \mathbf{q}_{W|R} \left| \begin{array}{l} \mathbf{q}_{W|R} \geq 0, \forall r \left( \sum_w q_{w|r} = 1 \right), q_{w_0|r_0} \neq q_{w_0|r_1}, \\ \mathbf{q}_{RX|c} \geq 0, \sum_{rx} q_{rx|c} = 1, \forall x \neq 0 \left( q_{r_0xc} = 0 \right), \\ \text{where } \mathbf{q}_{RX|c} = [\mathbf{q}_{W|R}]^{-1} \mathbf{p}_{WX|c}. \end{array} \right. \right\}. \tag{57}$$

We have already solved $\mathscr{B}_c$ before, it is the ZI MCAR compatibility set of $q(W \mid R)$ in Theorem 3. Then all we need is to take the intersection of these results, one for each $c$. This intersection $\mathscr{B} = \cap_{c \in \mathscr{C}} \mathscr{B}_c$ is non-empty, because there is some $q(R,X,W,C)$ produces the given observed law. First, the identification of $q_{w_0|r_1}$ and marginal constraints are

$$\forall c \in \mathscr{C}, \forall x \neq 0, q_{w_0|r_1} = p_{w_0|x_1,c} = p_{w_0|x,c}. \tag{58}$$

The last equality is due to the marginal constraint discussed in Theorem 3. Then we can write

$$q_{w_0|r_1} = p_{w_0|x_1}. \tag{59}$$

Next, we consider each case of the bound for $q_{w_0|r_0}$.

1. Suppose $p_{w_0|x_0,c'} > p_{w_0|x_1,c'}$ for some $c'$, then by Theorem 3, $q_{w_0|x_0} > p_{w_0|x_1,c'} = q_{w_0|r_1}$, where last equality follows from equation 58.

2. Suppose $p_{w_0|x_0,c''} < p_{w_0|x_1,c''}$ for some $c''$, then by Theorem 3, $q_{w_0|x_0} < p_{w_0|x_1,c''} = q_{w_0|r_1}$, where last equality follows from equation 58.

This means these 2 cases disjoint, i.e., we must have the following marginal constraint

$$\text{either } \forall c \left( p_{w_0|x_0,c} \leq p_{w_0|x_1} \right) \text{ or } \forall c \left( p_{w_0|x_0,c} \geq p_{w_0|x_1} \right). \tag{60}$$

The corresponding bounds are

$$q_{w_0|r_1} = p_{w_0|x_1}$$
$$q_{w_0|r_0} \in \begin{cases} [\max_c p_{w_0|x_0,c}, 1] & \text{if } \exists c', p_{w_0|x_0,c'} > p_{w_0|x_1}, \\ [0, \min_c p_{w_0|x_0,c}] & \text{if } \exists c', p_{w_0|x_0,c'} < p_{w_0|x_1}, \\ (0,1) \setminus \{p_{w_0|x_1}\} & \text{if } \forall c, 0 < p_{w_0|x_0,c} = p_{w_0|x_1} < 1. \end{cases}$$

The max/min appears since we take the intersection of the bounds for $c$. Moreover, if $\forall c, p_{w_0|x_0,c} = p_{w_0|x_1}$, then $\forall c, 0 < p_{w_0|x_0,c} < 1$ is an additional condition, and zero inflation does not occur, i.e., $q(R=0) = 0$. Due to the marginal constraints above, this exhausts all the cases.

Since the compatibility set $\mathscr{B}$ is the intersection of each $\mathscr{B}_c$, each is sharp in their own ZI MCAR model, the above abound is sharp.

$\square$

We collect the marginal constraints obtained from the proofs of Theorem 4 and Theorem 5 into the following lemma,

**Lemma 2.** *For a ZI MAR model in Fig. 2 (b) under $\mathbf{A1}^\dagger$ and $\mathbf{A2}^\dagger$, the observed law $p(X,W,C)$ obeys*

$$\forall c, \forall x \neq 0, p_{w_0|x,c} = p_{w_0|x_1,c}. \tag{61}$$

*For a ZI MAR model in Fig. 2 (c) under $\mathbf{A1}^*$ and $\mathbf{A2}^*$, the observed law $p(X,W,C)$ obeys*

$$\forall c, \forall x \neq 0, p_{w_0|x,c} = p_{w_0|x_1}, \tag{62}$$
$$\text{either } \forall c \left( p_{w_0|x_0,c} \leq p_{w_0|x_1} \right) \text{ or } \forall c \left( p_{w_0|x_0,c} \geq p_{w_0|x_1} \right).$$

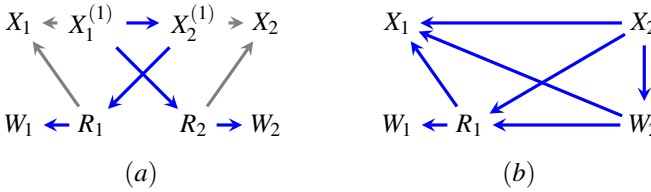

Figure 8: (a) The ZI bivariate block-parallel model. (b) The model Markov to this graph contains the marginal model for $p(X_1, X_2, R_1, W_1, W_2)$ in (a).

### A.6   ZI MNAR PROOFS

**Lemma 3.** *Consider any ZI model in Section 3.2.1 under* **A1**$^*$ *and* **A2**$^*$. *Denote* $Z_k \triangleq \{X, W, C\} \setminus \{W_k, X_k\}$. *The observed law* $p(X, W, C)$ *must satisfy, for each k,*

$$
\begin{cases}
\forall z_k, \forall x \neq 0, p_{w_k 0 | x_k = x, z_k} = p_{w_k 0 | z_{k1}}, \\
\forall z_k \left( p_{w_k 0 | x_{k0}, z_k} \leq p_{w_k 0 | x_{k1}} \right) \; or \; \forall z_k \left( p_{w_k 0 | x_{k0}, z_k} \geq p_{w_k 0 | z_{k1}} \right).
\end{cases}
\tag{63}
$$

*Proof.* Let $\tilde{\mathscr{G}}$ be the graph where $\{X_k, Z_k, R_k\}$ is fully connected, and $R_k \to W_k$. The Markov model $\mathscr{P}_{\tilde{\mathscr{G}}}$ for this graph contains all joint distributions $q(X_k, W_k, Z_k, R_k) = q(W_k \mid R_k) q(X_k, R_k, Z_k)$, where $q(X_k, R_k, Z_k)$ is from the saturated model restricted by **Z**. In the original ZI model, we have $p(X, W, C, R_k) = p(W_k \mid R_k) p(X_k, R_k, Z_k)$ with $p(X_k, R_k, Z_k)$ satisfying **Z**. Hence the model for this joint distribution is contained in $\mathscr{P}_{\tilde{\mathscr{G}}}$.

Subsequently, one could repeat the proof of Theorem 5 to find the bound for $q(W \mid R)$, with $W_k$ as $W$, $X_k$ as $X$, and $Z_k$ as $C$. The bound is not sharp, because we do not know the exact model for $q(X_k, R_k, Z_k)$. Similarly, application of lemma 2 yields the desired constraints. $\square$

## B   SIMULATIONS

### B.1   BOUND VALIDITY IN RANDOM DGPS

DGPs for ZI MCAR and ZI MAR are randomly selected according to Fig. 2 (a) and (b), respectively. In particular, a DGP for ZI MAR is a joint distribution which factorizes as

$$
p(X, X^{(1)}, R, W, C) = p(X \mid R, X^{(1)}) p(X^{(1)} \mid C) p(R \mid C) p(C) p(W \mid R)
\tag{64}
$$

Then, the observed law is $p(X, W, C) = \sum_{X^{(1)}, R} p(X, X^{(1)}, R, W, C)$.

We randomly select a DGP by sampling the following parameters

$$
\begin{aligned}
p(C = 0) &\sim \text{Uniform}[0, 1] \\
p(X^{(1)} = 0 \mid C = 0) &\sim \text{Uniform}[0, 1] \\
p(X^{(1)} = 0 \mid C = 1) &\sim \text{Uniform}[0, 1] \\
p(R = 0 \mid C = 0) &\sim \text{Uniform}[0, 1] \\
p(R = 0 \mid C = 1) &\sim \text{Uniform}[0, 1] \\
p(W = 0 \mid R = 0) &\sim \text{Uniform}[0, 1] \\
p(W = 0 \mid R = 1) &\sim \text{Uniform}[0, 1]
\end{aligned}
\tag{65}
$$

Further more, to satisfy the ZI-consistency

$$
p(X = 0 \mid R = 0, X^{(1)}) = 1; \quad p(X = 1 \mid R = 1, X^{(1)} = 1) = 1; \quad p(X = 0 \mid R = 1, X^{(1)} = 0) = 1.
\tag{66}
$$

## B.2 NUMERICAL BOUNDS RESULTS

We compute numerical bounds using method in Duarte et al. [2023] and compare to our analytical bounds for DGPs in ZI MCAR and ZI MAR. Since computation time for the dual bound may be very long (some DGP might take more than 36 hours), we report only DGPs where primary bound is available (whose computation time may take only a few minutes). We refer reader to original paper for distinction of dual/primal bounds.

| dgp | lb | ub | num lb | num ub | $p_{w0|r0}$ |
|---|---|---|---|---|---|
| 0 | 0.556406 | 1.0 | 0.556411 | 1.0 | 0.820732 |
| 1 | 0.357830 | 1.0 | 0.357830 | 1.0 | 0.493695 |
| 2 | 0.0 | 0.520689 | 0.0 | 0.520689 | 0.453609 |
| 4 | 0.606499 | 1.0 | 0.606499 | 1.0 | 0.682699 |
| 5 | 0.0 | 0.524069 | 0.0 | 0.524061 | 0.496676 |
| 6 | 0.381825 | 1.0 | 0.381825 | 1.0 | 0.441227 |
| 8 | 0.652288 | 1.0 | 0.652288 | 1.0 | 0.659347 |
| 9 | 0.698149 | 1.0 | 0.698149 | 1.0 | 0.738794 |
| 10 | 0.0 | 0.443595 | 0.0 | 0.443595 | 0.442502 |
| 11 | 0.656867 | 1.0 | 0.656867 | 1.0 | 0.850498 |
| 12 | 0.211359 | 1.0 | 0.211359 | 1.0 | 0.856658 |
| 14 | 0.183034 | 1.0 | 0.183034 | 1.0 | 0.303129 |
| 15 | 0.648430 | 1.0 | 0.648430 | 1.0 | 0.833933 |
| 16 | 0.292337 | 1.0 | 0.292337 | 1.0 | 0.307559 |
| 17 | 0.500542 | 1.0 | 0.500542 | 1.0 | 0.553972 |
| 18 | 0.0 | 0.102988 | 0.0 | 0.102988 | 0.087253 |
| 20 | 0.0 | 0.479532 | 0.0 | 0.479532 | 0.238318 |
| 21 | 0.426615 | 1.0 | 0.426615 | 1.0 | 0.426787 |
| 22 | 0.399169 | 1.0 | 0.399169 | 1.0 | 0.494816 |
| 23 | 0.0 | 0.216052 | 0.0 | 0.216052 | 0.158163 |
| 24 | 0.436636 | 1.0 | 0.436636 | 1.0 | 0.533412 |
| 26 | 0.429579 | 1.0 | 0.429579 | 1.0 | 0.710488 |
| 27 | 0.0 | 0.500198 | 0.0 | 0.500199 | 0.451856 |
| 28 | 0.0 | 0.383471 | 0.0 | 0.383471 | 0.136093 |
| 29 | 0.0 | 0.325871 | 0.0 | 0.325871 | 0.070747 |
| 30 | 0.363744 | 1.0 | 0.363744 | 1.0 | 0.374293 |

Table 2: Comparison between our analytical lower and upper bound (*lb*/*ub*) to numerical bounds (*num lb*/*num ub*) for a randomly selected set of DGPs in ZI MCAR model corresponding to Fig. 2 (a) (reproduced in Fig. 5). True $p_{w_0|r_0}$ is reported.

| dgp | lb | ub | num lb | num ub | $p_{w0|r0}$ |
|---|---|---|---|---|---|
| 0 | 0.0 | 0.429089 | 0.0 | 0.429089 | 0.413267 |
| 1 | 0.834644 | 1.0 | 0.834644 | 1.0 | 0.848638 |
| 2 | 0.0 | 0.340484 | 0.0 | 0.340484 | 0.319264 |
| 3 | 0.300217 | 1.0 | 0.300217 | 1.0 | 0.515513 |
| 4 | 0.582249 | 1.0 | 0.582249 | 1.0 | 0.688620 |
| 5 | 0.938604 | 1.0 | 0.938604 | 1.0 | 0.991572 |
| 6 | 0.0 | 0.147758 | 0.0 | 0.147758 | 0.053637 |
| 7 | 0.534321 | 1.0 | 0.534321 | 1.0 | 0.569545 |
| 8 | 0.720775 | 1.0 | 0.720775 | 1.0 | 0.726467 |
| 9 | 0.585611 | 1.0 | 0.592962 | 1.0 | 0.686385 |
| 10 | 0.261442 | 1.0 | 0.261442 | 1.0 | 0.303129 |
| 11 | 0.378136 | 1.0 | 0.378136 | 1.0 | 0.481036 |
| 12 | 0.0 | 0.729282 | 0.0 | 0.729282 | 0.703234 |
| 13 | 0.425249 | 1.0 | 0.425249 | 1.0 | 0.426797 |
| 14 | 0.612665 | 1.0 | 0.612665 | 1.0 | 0.632688 |
| 15 | 0.319180 | 1.0 | 0.319180 | 1.0 | 0.628988 |
| 16 | 0.0 | 0.702582 | 0.0 | 0.702582 | 0.660187 |
| 17 | 0.661849 | 1.0 | 0.661849 | 1.0 | 0.726963 |
| 18 | 0.594456 | 1.0 | 0.594456 | 1.0 | 0.600720 |
| 19 | 0.531541 | 1.0 | 0.532509 | 1.0 | 0.536156 |
| 21 | 0.596110 | 1.0 | 0.600331 | 1.0 | 0.606306 |
| 22 | 0.513144 | 1.0 | 0.513144 | 1.0 | 0.692834 |
| 23 | 0.0 | 0.560519 | 0.0 | 0.560519 | 0.536384 |
| 24 | 0.837194 | 1.0 | 0.837212 | 1.0 | 0.844800 |
| 25 | 0.0 | 0.443658 | 0.0 | 0.443658 | 0.302563 |
| 26 | 0.469323 | 1.0 | 0.469323 | 1.0 | 0.479800 |
| 28 | 0.688084 | 1.0 | 0.688084 | 1.0 | 0.826820 |
| 30 | 0.0 | 0.230720 | 0.0 | 0.230720 | 0.103594 |

Table 3: Comparison between our analytical lower and upper bound (*lb/ub*) to numerical bounds (*num lb/num ub*) for a randomly selected set of DGPs in ZI MAR model corresponding to Fig. 2 (c) (reproduced in Fig. 7). True $p_{w_0|r_0}$ is reported.

# C  VARIABLE DESCRIPTIONS IN THE CLABSI DATA APPLICATION

This section describes the covariates used in the CLABSI data application (all coded as binary variables). These covariates correspond to types of therapy, and types of catheter used.

- `Pediatrics`: the CVC therapy is tailored for children.
- `Chemotherapy`: the CVC therapy is used to administer chemotherapy.
- `OPAT`: outpatient parenteral antimicrobial therapy (IV antibiotics).
- `TPN`: parenteral nutrition delivered via the VC.
- `Other therapy`: any other type of therapy not included in the above categories, such as hydration.
- `Port`: a type of CVC in use.
- `PICC`: peripherally inserted central catheter, another type of CVC.
- `Tunneled CVC`: a CVC tunneled under the skin.