# OpenReview forum: "Zero Inflation as a Missing Data Problem: a Proxy-based Approach"
_auai.org/UAI/2024/Conference — UAI 2024 poster_

### Official Review · Reviewer_QAXB · 2024-02-27

**Q2-1 Originality-Novelty:** 3
**Q2-2 Correctness-Technical Quality:** 2
**Q2-5 Clarity Of Writing:** 2

**Q10 Ethical Concerns:**

No.

**Q1 Summary And Contributions:**

Zero-inflated data is a common problem in real-world datasets. Without significant assumptions, identification is not possible in such datasets, and making unbiased causal estimates is challenging. The authors suggest using graphical models to represent the causes of zero-inflation. Under this representation, the authors propose using proxies of missingness indicators to identify target parameters. When the proxy-indicator relationship is not known, the authors pose a sensitivity analysis with bounds for target parameters. The theoretical results are confirmed by synthetic and non-synthetic experiments.

**Q2-3 Extent To Which Claims Are Supported By Evidence:**

3: Good: the main claims are supported by convincing evidence (in the form of adequate experimental evaluation, proofs, (pseudo-)code, references, assumptions).

**Q2-4 Reproducibility:**

2: Fair: key resources (e.g. proofs, code, data) are unavailable but key details (e.g. proof sketches, experimental setup) are sufficiently well-described for an expert to confidently reproduce the main results.

**Q3 Main Strengths:**

Zero-inflation is a common problem across the empirical and social sciences, and this work presents a solid addition to the current literature that practitioners can draw upon to reduce biases in their analyses. I believe the main strengths of this work are as follows:

1. This paper presents a very comprehensive and honest theoretical analysis. The authors provide proofs for identification of the full law and are transparent about the impracticability of the identification strategy due to the difficulty of obtaining the distribution p(W | R). To combat this, the authors discuss a partial identification strategy that gives bounds on target parameters. Finally, the authors provide fully synthetic simulations and a real-world application case study.

2. The authors in this paper present a novel solution for overcoming zero-inflation that creatively combines graphical approaches to missing data and single-proxy proximal causal learning. The creativity and originality is impressive. In addition, the authors’ full law identification argument is clear and easy to follow.

3. The data application in section 4.3 presents promising results and a plausible situation under which the assumptions for the authors’ proposed method are satisfied. I feel convinced that the proposed method can mitigate undercounting in zero-inflated datasets. Specifically, I think adjudicator access to an electronic health records system is a reasonable proxy variable for the missingness indicator.

Great work!

**Q4 Main Weakness:**

I believe the main weaknesses of this work are as follows:

1. My main critique of this paper is that the authors do not provide a method for practitioners to verify whether assumption (A1) and its variations are fulfilled. I believe practitioners may find the proposed method more useful if they can confidently verify or guarantee whether key assumptions for the method are fulfilled. I believe this paper can also benefit from a discussion of possible proxies W that can plausibly appear in real-world datasets.

2. It may be difficult for other scientists to reproduce the data application described in section 4.3. Is there a specific reason why the case analysis code and data are not available? In addition, is it possible to be more specific about where the data for patients undergoing CVC treatments is from? It is understandable if the datasets cannot be accessed by the public, but I believe it is important to communicate this information in order to increase the reproducibility of the authors’ work.

3. This could be a misunderstanding on my part of the paper, but the authors mention multiple times that the missingness indicator R may be unobserved whenever the observed proxy X=0, but section 2.2’s definition of the zero-inflation problem gives only two possible values for the missingness indicator R (is it possible for R to also have a value of ‘?’ ?). Furthermore, it is not clearly stated in section 2.2 (although it is stated elsewhere) when R is unobserved. I believe that the behavior of the missingness indicator R can be more clearly defined relative to the counterfactual variable X(1), observed proxy X, and proxy variable W. In the missing data literature, authors sometimes define the relationship between X(1) and X using a piecewise function; such a way of defining the relationships between variables may make the rest of the paper clearer for the reader.

**Q5 Detailed Comments To The Authors:**

In section 2.2, is the relationship “X(1)_i = 0 if R_i = 0” the correct one? It does not seem right to me that the value of the counterfactual depends on the missingness indicator. Did the authors mean that X_i = 0 if R_i = 0? If this is a typo, fixing it should make the rest of the paper easier to understand.

In your DAGs on Figure 2, you have R causing a proxy W. Could you please specify and discuss some real-world situations where this might occur?

In section 4.3, the authors refer back to Figure 2(b), but I do not believe that the DAG accurately describes the DGP for the data application; W should cause R instead of the other way around. Perhaps this setup for the DAG is more realistic in real-world datasets as well.

In Table 1, the bounds are quite big (for example, the bound for row 1 in MCAR is 0.3578 to 1.0). Is there a reason why the lower bound is often 0.0 and the upper bound is often 1.0?

In section 4.1, can you also make sure that your bounds do not cover the true p(W | R) when assumptions are violated in the fully synthetic DGP?

In paragraph 2 of section 4.3, is prevance a typo in the following sentence? “In fact, the observed CLABSI rate in our data was more than 65%, much higher than the prevance in the population undergoing CVC therapies”?

In the light of author discussions and rebuttals, I have decided to change my score to a weak accept.

**Q9 Complying With Reviewing Instructions:**

Yes

---

> ### Author Rebuttal · Authors · 2024-04-05
>
> We thank the reviewer for their insightful comments and their encouragement.
>
> We consider the first point raised by the reviewer. We would like to thank them for this thoughtful comment. Like most missing data models, our assumptions are not testable.  However, because our model imposes restrictions on the observed data distribution, it is falsifiable (for example, lemma 2 and 3).  In other words, if restrictions imposed by our model do not hold in the observed data, we can rule out our model.  However, if the restrictions do hold, our model may or may not hold.  The situation is analogous to causal models such as the instrumental variable model, which is similarly falsifiable (but not testable) via violations of the instrumental inequality, or the Bell model with classical hidden variables in quantum mechanics, which has been falsified via violations of the Bell inequality.  We would be happy to add more discussion of this important point.
>
> Although our theorems are expressed for binary variable $X^{(1)}$, it is straight-forward to extend the results to categorical variable $X^{(1)}$, in which the bounds are similar. Additionally, even in ZI MCAR, there is a marginal constraint, which could also be used as a falsification test.
>
> We thank the reviewer for the suggestion on the plausible indicator proxies $W$ in real data. We find that as has been observed in the proximal causal inference literature, informative proxies are available in many practical cases. We would incorporate this suggestion into our updated version of the paper.
>
> We consider the second point raised by the reviewer. We would be happy to share all data processing and analysis code, as well as all simulation studies. However, our data application involves real patient data from a set of institutions. While we can publish results of analyses on such data, publishing patient data itself would violate patient confidentiality, and is thus not allowed by the IRB application approved by our institution. We would be happy to describe the origin of the data in detail if the paper is accepted.  We did not include these details in our submission to preserve blinding.
>
> We consider the third point raised by the reviewer. We would like to clarify the zero-inflation consistency. The indicator $R$ is a binary variable taking 0 or 1 as values. The counterfactual variable $X^{(1)}$ is categorical and could take value in $\\{0, 1, …, N\\}$ (although much of our current paper describes binary $X^{(1)}$). Then the observed variable $X$ is defined as follow: $X = I(R=0) * 0 + I(R=1) * X^{(1)}$. In other words, $X = X^{(1)}$ when $R=1$, and $X = 0$ when $R=0$. From this definition, if we observe a non-zero value for $X$, it is certainly the case that $R=1$. On the other hand, if $X=0$, we don’t know if $R=1$ or $R=0$. Therefore, $R$ is considered partially observed.
>
> Furthermore, no variable in a zero inflation problem we consider takes “?” as a value. This is to be compared to missing data problem, in which $R$ is also binary, $X^{(1)}$ is categorical, and $X$ is defined as $X = I(R=0) * ? + I(R=1) * X^{(1)}$. That is, $X = X^{(1)}$ when $R=1$, and $X = ?$ when $R=0$.
>
> We consider comment 1 by the reviewer. The relationship “$X^{(1)}_i = 0$ if $R_i = 0$” is not a correct one. Thank you for pointing out this error. We indeed meant $X = X^{(1)}$ when $R=1$, and $X = 0$ when $R=0$ as the zero-inflation consistency. We apologize for the confusion.
>
> We consider comment 2 and 3 by the reviewer. In our data application, $R$ represents ‘true state of knowledge of the adjudicator’ and $W$ represents ‘a possibly imperfect recording of the true state of knowledge of the adjudicator, as represented by whether they were granted access to details about the patient’s case via electronic health records.’ In this sense, $R$ is the ‘true variable’ we don’t always see, and $W$ is an ‘observed proxy.’  Thus, it makes sense for $R$ to influence $W$. However, we can imagine situations where $W$ would cause $R$, and we may still be able to obtain identification. This is indeed the case for figure 2(c), where flipping the edge $R \rightarrow W$ to $W \rightarrow R$ would result in the same model, which is described in theorem 4. We performed analysis using this model and reported the result in our data application section 4.3.

---

### Official Review · Reviewer_7yz3 · 2024-02-29

**Q2-1 Originality-Novelty:** 3
**Q2-2 Correctness-Technical Quality:** 3
**Q2-5 Clarity Of Writing:** 4

**Q1 Summary And Contributions:**

The paper considers zero-inflation as a missing data problem and proposes a strategy for full law identification using proxy variables. Partial identification and bounds are also studied.

**Q2-3 Extent To Which Claims Are Supported By Evidence:**

4: Excellent: all claims are supported by very convincing evidence (in the form of comprehensive experimental evaluation, rigorous mathematical proofs, detailed (pseudo-)code, precise references, well-motivated and realistic assumptions) and the authors deliver what they promise.

**Q2-4 Reproducibility:**

4: Excellent: key resources (e.g. proofs, code, data) are available and key details (e.g. proof sketches, experimental setup) are comprehensively described for competent researchers to confidently and easily reproduce the main results.

**Q3 Main Strengths:**

The problem of observations incorrectly labeled as zero is important and this paper provides nonparametric solutions that are more general than model-based parametric approaches.

The assumptions related to the main theorems are clearly stated.

Partial identification and bounds are studied both formally and validated empirically using simulations.

There is a suitable motivating real data application.

**Q4 Main Weakness:**

The paper is notationally heavy, which is common in identification theory and not really a fault of the authors.

The paper considers only binary variables.

**Q5 Detailed Comments To The Authors:**

I disageree with the authors' use of the term "zero inflated data". In typical statistical literature, ZI is considered to arise from a DGP with two processes, one that generates zeros and one that generates counts that can be zero (a zero inflated probability distribution). The authors also mention this as an approach to ZI data modeling, but such processes can and often do arise in reality. The kind of data the authors consider, where missing value is incorrectly labeled as zero, is not ZI in this sense, it is simply incorrectly recorded data. Can the authors provide some references where the term ZI is used with the same interpretation as in this paper? Otherwise, I think a different term should be used, as it can be misleading.

Suppose that we can point identify the full law using Theorem 1 (or Theorem 2). I would imagine that using Eq. 2 (or Eq. 3) in practice does not always result in valid probabilities when the right-hand side terms are estimated from the data because the formula contains an inverse matrix of probabilities. I think this should be mentioned.

The authors mention and use the autobounds package. Is it straightforward to convert a ZI missing data problem instance into a corresponding program and could this be automated with an easy-to-use interface? The code in the supplementary material does not seem overly complicated at least.

Missing data identification theory often considers binary variables as is done here. Does the proxy approach presented here naturally extend to categorical variables (either for the proxy or the response of interest)?

Minor comments:

What is the difference between the gray and blue edges in the figures? It seems that gray edges are used to denote deterministic connections, but in Figure 2, there are blue edges X^{(1)} -> X and R -> X. Should these also be gray? The colors should be explained in the text.

MCAR, MAR and MNAR missingness types should be explained in the introduction.

When introducing missing data DAGs, the term m-DAG should be introduced, e.g., "Missing data DAGs (m-DAG) ..."

The reference Bhattacharya et al. 2020 has wrong author order, it should be Nabi et al. (Full law identification in graphical models of missing data: Completeness results).

Section 4.3, second paragraph on page 8: prevance -> prevalence

Section 4.3, last paragraph on page 8: cetheter -> catheter

**Q9 Complying With Reviewing Instructions:**

Yes

---

> ### Author Rebuttal · Authors · 2024-04-05
>
> We thank the reviewer for the thoughtful comments.
>
> We consider the first point raised by the reviewer. We note that the results of our paper extend to categorical variables, with similar results and bounds.  We have derived these results already and would be happy to include them.
>
> We consider the second point raised by the reviewer. We can certainly view our proposed setup as a measurement error generalization of missing data, for example, and we would be happy to point that out. However, we believe viewing our setup as zero inflation is fair, because: (a) we have more zeros than there should be, and (b) there are, in fact, two processes: one that generates counts (e.g. true outcome values), and another that generates excess zeros for some true outcome values. The former is the process corresponding to the full data distribution, and the latter is the censoring process resembling those found in missing data. Thus, we view our setup as a `missing data like’ zero inflation setup.  What’s more, we think setups of the sort we consider indeed arise in practice, as we believe our CLABSI example shows. We certainly don’t mean to invalidate the existing rich literature on zero inflated data, which is appropriate in other contexts.
>
> We consider the third point raised by the reviewer on the matrix inverse equation. Thank you for this insightful question. This is indeed the motivation of the compatibility issue our method tries to address. The reason matrix inversion equation gives an invalid probability $p(R,X,C)$ as output is that $p(W|R)$ and $p(X, W, C)$ may not be compatible. In other words, no joint distribution $p(R,W,X,C)$ in the model yields these $p(W|R)$ and $p(X, W, C)$. The Kuroki-Pearl method requires both $p(W|R)$ and $p(X, W, C)$ to be the ground -truth, i.e., there is a joint distribution producing them. This implicitly solves the compatibility condition for their setting, but yields no guidance on how to achieve it in practical problems. Providing this guidance for our setting is one of our contributions.
>
> In practice, we rarely know the ground truth for $p(W|R)$ or have access to  data to estimate this object, because $R$ is partially unobserved. Similarly, we don’t know the ground truth for $p(X,W,C)$ either, and must estimate $\hat{p}(X, W, C)$ from data. Thus, it is crucial that we make sure that our guess $p(W|R)$ and the estimated $\hat{p}(X,W,C)$ are compatible.
>
> To illustrate this incompatibility issue, we run the following additional simulation
> We create a DGP (a Bayesian network) w.r.t. ZI MCAR graph in Figure 2(a), and obtain the ground-truth $p(W|R)$. We then sample $100000$ data points $(W_i, X_i)$ from this DGP, and estimate $\hat{p}(W,X)$ by counting (non-parametric MLE for categorical data). Finally, we calculate $p(R,X)$ using the Kuroki-Pearl matrix inversion equation. The estimated $p(R,X)$ has negative elements, rendering it invalid.
> ```
> [ 0.5178 -0.0830]
> [ 0.0558  0.5092]
> ```
> Our method addresses the compatibility problem by providing bounds for $p(W|R)$ w.r.t. a given $p(X, W, C)$. Any $p(W|R)$ inside the bounds is compatible, in the sense that (i) the inversion equation gives valid probability $p(R,X,W,C)$, because (ii) the chosen $p(W|R)$ and $p(X, W, C)$ actually belong to some joint distribution in the model.
>
> Our simulations represent a plausible real-world situation where we don’t know the ground-truth $p(W|R)$, yet we may have (infinite) data for $(X, W, C)$, i.e., the ground-truth $p(X,W,C)$. The simulations show that any $p(W|R)$ inside the bounds is indeed compatible with this given $p(X, W, C)$. We also show that the ground-truth $p(W|R)$ lies in this bound.
>
> Our data application shows how the bound could be applied in practice. We first estimate $p(X,W,C)$ using the EM algorithm. Then we run our methods (theorem 4, 5) on this estimated $p(X,W,C)$, producing a compatible bounds for $p(W|R)$. We grid search the bounds, computing $p(R, X, W, C)$ for each $p(W|R)$, then recover the underlying full law $p(X^{(1)}, R, W, C)$. The mean $E[X(1)]$ is our final target of estimation.
>
> We consider the fourth point raised by the reviewer. We believe the procedure for encoding restrictions in DAG models of ZI data we considered in the autobounds package is fairly straightforward.  However, we have found that details of encoding the problem in autobounds matter a great deal for efficiency of the resulting polynomial program.
>
> For example, the ZI consistency could be encoded in 2 ways
> - $p(X=x | R=1, X(1)=x) = 1$ and $p(X=0 | R=0, X(1)=x) = 1$ , or
> - $p(X \neq X1, R=1, C, W) = 0$ and $p(X \neq 0, X1, R=0, C, W) = 0$.
>
> The second encoding seems to result in a much more efficient polynomial program.
>
> Fortunately, the authors made the autobounds freely available, and were able to provide useful advice on "encoding tricks."  We would be happy to provide guidance to other practitioners on how to accomplish effective encodings in the supplement.

---

### Official Review · Reviewer_UqUY · 2024-03-22

**Q2-1 Originality-Novelty:** 3
**Q2-2 Correctness-Technical Quality:** 3
**Q2-5 Clarity Of Writing:** 3

**Q1 Summary And Contributions:**

This paper tackles the problem of full law identification with zero-inflated data.

Zero-inflated data can be viewed as data with missingness while the missingness indicator is partially latent, i.e., when the observed variable Xi=0, it's unknown whether Ri is 1 or 0, i.e., whether the observed zero is true or error. This is in a way related to self-censoring mechanisms.

Therefore in this paper, the authors assumes the availability of a proxy variable Wi for missingness indicator Ri, s.t., given Ri, this proxy Wi is conditionally independent of all the true data and other variables' missingness indicators.

The identification strategy then consists of two steps: 1) indicator restoration and 2) downstream identification. The 2nd step is standard and this paper focuses on the 1st step. Identification methods and results are given under different missingness mechanisms, basically with the tool of algebra properties of the probability transition matrix to adjust for the bias caused by unobserveness. Bounds are also given.

**Q2-3 Extent To Which Claims Are Supported By Evidence:**

3: Good: the main claims are supported by convincing evidence (in the form of adequate experimental evaluation, proofs, (pseudo-)code, references, assumptions).

**Q2-4 Reproducibility:**

2: Fair: key resources (e.g. proofs, code, data) are unavailable but key details (e.g. proof sketches, experimental setup) are sufficiently well-described for an expert to confidently reproduce the main results.

**Q3 Main Strengths:**

1. Significance. The problem tackled here is highly significant, especially in scRNA-seq data. Most existing methods in dealing with inflated zeros rely on restricted parametric assumptions or unidentifiable imputation methods. This paper, by offering a more generalized, nonparametric approach, addresses a critical gap. To this end, a recent paper https://openreview.net/forum?id=gFR4QwK53h can be discussed here with some observations similar, while their objective is purely on structure learning.

2. Technical novelty and comprehensiveness. The motivation behind this work seems straightforward: though Ri is not directly observable, we introduce a Wi that carries only information of Ri to separate its information out. However, in exact identification, even if p(W|R) is known the identification is not exactly the same as that in standard missingness problem, and as for tools, it's not trivially using rank regularity constraints as in existing work. Partial identification results are given in difference cases, and bounds are also given.

**Q4 Main Weakness:**

1. More justifications are needed for the assumed availability of the proxies for missingness indicators. In real-world datasets (especially for scRNA-seq data), how to reliably find/design/justify such proxies? It requires that we almost already have missingness indicators, though as a noisy measurement to them. Also, are assumptions A0, A1*, and A2* testable?

2. By using the rank regularity constraints on probability transition matrices, the data needs to be discrete, which is another strong limitation for real-world applicabilities. Can this be relaxed, in a way like https://arxiv.org/abs/2011.04504 w.r.t. https://arxiv.org/abs/1609.08816?

3. A discussion with existing works on self-censoring missingness mechanisms (e.g., https://openreview.net/pdf?id=DrGiy3l4r27) is expected, as when the missingness indicators are partially latent, an analogy can be drawn between self-censored data and zero-inflated data, even when the missingness happens CAR.

4. Minor, on clarity: in Section 2.1 line 5, the observed variables X is called "proxies". This should be differentiated between the later usage of "proxies" on W.

**Q5 Detailed Comments To The Authors:**

See above "weaknesses" part.

**Q9 Complying With Reviewing Instructions:**

Yes

---

> ### Author Rebuttal · Authors · 2024-04-05
>
> We would like to thank the reviewer for the careful review and thoughtful comments.
>
> Firstly, we thank the reviewer for the suggested paper. Indeed, this paper concerns with the causal discovery problem. We would be happy to add more discussion of related work on gene regulatory networks, another common application area where zero inflation arises.
>
> We consider the first weakness raised by the reviewer. Like most missing data models, our assumptions are not testable.  However, because our model imposes restrictions on the observed data distribution (for example, those given by lemmas 2 and 3), it is falsifiable.  In other words, if restrictions imposed by our model do not hold in the observed data, we can rule out our model.  However, if the restrictions do hold, our model may or may not hold.  The situation is analogous to causal models such as the instrumental variable model, which is similarly falsifiable (but not testable) via violations of the instrumental inequality, or the Bell model with classical hidden variables in quantum mechanics, which has been falsified via violations of the Bell inequality.  We would be happy to add more discussion of this important point.
>
> We consider the second weakness raised by the reviewer. Thank you for this insightful question. A generalization of our approach to real-valued variables by extending proximal causal inference methods is a very interesting area of future work but, we feel, beyond the scope of this paper.  We note that by their nature, missingness indicators are binary, which simplifies finding proxies. This means that our approach avoids conceptual difficulties the original Kuroki-Pearl paper ran into (since in practice unobserved confounders are often real-valued vectors, not categorical variables, as required by the Kuroki-Pearl matrix inversion approach).  However, if other variables in the problem are real-valued, matrix inversion must be replaced by finding solutions of Fredholm equations of the first kind.  Note also that if the zero inflated outcome is real-valued, the problem only makes sense in settings where the zero value has support.
>
> We consider the third weakness raised by the reviewer. We would be happy to include a comparison of zero inflated models with non-identified missing data models with fully observed indicators, such as the self-censoring model. Both models are not identified, and proxies may be used to recover identifiability, however the precise approach naturally differs, since the reason behind non-identification of models differs. Moreover, in missing data, the indicator is always observed, unlike our situation.
>
> Finally, we thank the reviewer for the excellent suggestion on the terms “proxies”. We would make this clearer.

---

### Official Review · Reviewer_nfD1 · 2024-03-22

**Q2-1 Originality-Novelty:** 3
**Q2-2 Correctness-Technical Quality:** 3
**Q2-5 Clarity Of Writing:** 2

**Q10 Ethical Concerns:**

No, but It seems like there is no section discussing the potential negative societal impact of the work.

**Q1 Summary And Contributions:**

The paper proposes a zero-inflated inference method, treating it as a missing data problem. The authors show that the full law is identifiable under certain conditions in an m-DAG given the proxy-indicator relationship. The proxy-indicator relationship is partially identifiable in certain ZI models.

**Q2-3 Extent To Which Claims Are Supported By Evidence:**

2: Fair: the main claims are somewhat supported by evidence (but the experimental evaluation may be weak, or does not match entirely with the claims, important baselines may be missing, proofs contain important ideas but lack rigor, algorithmic details are only discussed superficially, references are imprecise, assumptions are not sufficiently motivated or explicated, etc.).

**Q2-4 Reproducibility:**

3: Good: key resources (e.g. proofs, code, data) are available and key details (e.g. proofs, experimental setup) are sufficiently well-described for competent researchers to confidently reproduce the main results.

**Q3 Main Strengths:**

1. The authors conduct experiments on both simulated and real data.
2. The proof is in detail, and the code is provided.
3. The framework of the paper is clear.
4. The contributions of the paper are listed clearly.

**Q4 Main Weakness:**

1. The simulations are not about how to get the true realizations of the random variables from the inflated zero value but are to support one of the stages needed to achieve the final goal.
2. There is no baseline comparison, so measuring the method's performance and contributions is difficult.
3. Considering all the difficulties mentioned in the paper, is it worth treating zero-inflated data as a general type of missing data problem? What is the unique advantage of the proposed method?
4. From section 2, it seems that the random variable is constrained to be a binary variable. If this is true, the entire paper is about the binary variable. Is it better to express this in the paper's title or highlight this constraint more clearly?
5. There is no related work section. Is the proposed algorithm the first to treat the zero-inflation as a missing data problem?

**Q5 Detailed Comments To The Authors:**

1. Could you specify the assumptions needed for the second type of approach in the introduction section to show that they are indeed unlikely to hold in practice?
2. What is $W$ in the second paragraph in section 2?
3. Could you explain what you mean by “due to consistency” in section 2?
4. Could you introduce the full name of MAR and MNAR, and cite the related paper before using these phrases in the caption of Figure 1?
5. There is no $R_2$ in Figure 1 (b). If $R_2$ in the caption denotes $R_1$ in Figure 1 (b), as a collider, $X^{(1)}_1$ and $R_1$ should be d-connected conditioning on $C$, not d-separated.
6. In (A0), why $X_i=1$? Is $X^{(1)}_i$ already constrained to be a binary variable from section 2.2?
7. In section 3.1.1, should “$p(W|R)$ is known” and “$P(R, X, W, C)$ is given” be two additional assumptions needed in the Proposition 1? If so, is it better to have these two constraints listed as $A3$ and $A4$ in this section? Furthermore, are they too strong in general cases if they are two assumptions?
8. Please point out if I missed that part, but where is $A0^{*}$?
9. Is your final goal to identify $p(R, X^{(1)}, W, C)$ or $p(X^{(1)})$? How to identify the latter after identifying $p(R, X^{(1)}, W, C)$? It gave me the impression that the final goal of this zero inflation problem has not been achieved, and the paper is about to solve part of the stages. It would be great if you could summarize briefly here the whole process to obtain $p(X^{(1)})$ from a missing data problem, and which stages of the problem have been solved by this paper, and which parts can be solved by other existing methods.
10. Is it better to add some baselines in the real data case to show the advantage of applying the proposed method?
11. What is the ground truth of the CLABSI rate? If there is no ground truth, how do you know the obtained result is correct?

**Q9 Complying With Reviewing Instructions:**

Yes

---

> ### Author Rebuttal · Authors · 2024-04-05
>
> We thank the reviewer for their detailed reading of our paper and insightful questions.
>
> We would like to address the first weakness raised by the reviewer. The simulations are chosen to highlight our findings, rather than getting the true realisations of the random variables. As shown in theorem 1, 2, if we know the ground truth $p(W|R)$ and ground truth $p(X, W, C)$, we could recover the ground truth $p(R, X, W, C)$ by a modification of the Kuroki-Pearl effect restoration method, then recover the full law of the underlying variable $p(X(1), R, W, C)$ using proposition 1. We did not perform simulations to confirm this result, since the requirement that $p(W|R)$ is known is rather unrealistic. We would be happy to add them to clarify our method, however.
>
> In practice, we rarely know the ground truth for $p(W|R)$ or have access to  data to estimate this object, because $R$ is partially unobserved.  Similarly, we don’t know the ground truth for $p(X,W,C)$ either, and must estimate $\hat{p}(X, W, C)$ from data. Thus, it is crucial that we make sure that our guess $p(W|R)$ and the estimated $\hat{p}(X,W,C)$ are compatible. Compatibility simply means there is some joint distribution $p(R,W,X,C)$ in the model that yields these $p(W|R)$ and $p(X, W, C)$. Without compatibility, the matrix inversion in theorem 1 and 2 will result in invalid probability distribution $p(R,X,C)$.
>
> To illustrate this point, we run the following additional simulation:
> We create a DGP (a Bayesian network) w.r.t. ZI MCAR graph in Figure 2(a), and obtain the ground-truth $p(W|R)$. We then sample $100000$ data points $(W_i, X_i)$ from this DGP, and estimate $\hat{p}(W,X)$ by counting (non-parametric MLE for categorical data). Finally, we calculate $p(R,X)$ using the Kuroki-Pearl matrix inversion equation. The estimated $p(R,X)$ has negative elements, rendering it invalid.
> ```
> [0.5178 -0.0830]
> [0.0558  0.5092]
> ```
> This situation occurs because the non-parametric MLE for $p(W,X)$, namely counting realizations, places us outside the model.  Another way for this situation to arise is if the provided distribution $p(W|R)$ is not compatible with the combination of modeling assumptions and observed data.
>
> Our method addresses the compatibility problem by providing bounds for $p(W|R)$ w.r.t. a given $p(X, W, C)$. Any $p(W|R)$ inside the bounds is compatible, in the sense that (i) the inversion equation gives valid probability $p(R,X,W,C)$, because (ii) the chosen $p(W|R)$ and $p(X, W, C)$ actually belong to some joint distribution in the model. The Kuroki-Pearl method requires both $p(W|R)$ and $p(X, W, C)$ to be the ground-truth, i.e., there is a joint distribution producing them. This implicitly solves the compatibility condition for their setting, but yields no guidance on how to achieve it in practical problems.  Providing this guidance for our setting is one of our contributions.
>
> Our simulations represent a plausible real-world situation where we don’t know the ground-truth $p(W|R)$, yet we may have (infinite) data for $(X, W, C)$. The simulations are to show that any $p(W|R)$ inside the bounds is indeed compatible with this given $p(X, W, C)$. We also show that the ground-truth $p(W|R)$ lies in this bound.
>
> We consider the second weakness on the benchmark baseline. In the CLABSI data application, the natural baseline comparison is the assumption all zeros are real under a missing data model such as MAR, which is a result we report.  We also want to add that while we cannot directly validate our findings on the CLABSI data, since the ground truth is not directly available, we have consulted are clinical collaborators, who deemed our recovered bounds for the CLABSI rate reasonable, and consistent with current clinical knowledge for these types of infection in the population considered in our analysis.
>
> We would like to address the third weakness raised by the reviewer. We agree that the problem setup is more complicated than standard inflated zero setups, and thus requires the presence of informative proxies, and partial identification methods, as well as dealing with the model compatibility problem.
>
> However, we think in the types of situations such as our CLABSI application, the zero inflated version of missing data corresponds more closely to the type of substantive question of interest to the practitioner.  In particular, in our applications zeros are, in fact, incorrectly recorded in settings where a definite CLABSI determination is not possible, (rather than a missing value token such as ? and NA).  This naturally leads to the worry that CLABSI is undercounted in data recorded in this way.  This yields a methodological problem of determining which recorded zeros are ‘real’ and which are ‘fake,’ which led us to develop our method. In this setting, a mixture model for inflated zeros simply isn’t capturing the data generating process by which inflated zeros come about, nor do such models allow us to answer the relevant substantive question.

---

### Official Review · Reviewer_JHWM · 2024-03-23

**Q2-1 Originality-Novelty:** 2
**Q2-2 Correctness-Technical Quality:** 3
**Q2-5 Clarity Of Writing:** 2

**Q1 Summary And Contributions:**

This paper studies the inference for data problem with zero inflation, namely by imputing some zeros by censored realizations for distinguishing the true zeros and the artifical zeros. For this, they consider different missing data models, as Missing Completely At Random (MCAR) for instance, for establishing useful graphical models to have a proxy-based identification (of the zeros in particular). Some experiments are given for confirming their analytic results and an application to the CLABSI data is proposed.

**Q2-3 Extent To Which Claims Are Supported By Evidence:**

2: Fair: the main claims are somewhat supported by evidence (but the experimental evaluation may be weak, or does not match entirely with the claims, important baselines may be missing, proofs contain important ideas but lack rigor, algorithmic details are only discussed superficially, references are imprecise, assumptions are not sufficiently motivated or explicated, etc.).

**Q2-4 Reproducibility:**

2: Fair: key resources (e.g. proofs, code, data) are unavailable but key details (e.g. proof sketches, experimental setup) are sufficiently well-described for an expert to confidently reproduce the main results.

**Q3 Main Strengths:**

See zero-inflated data as a general type of missing data problem is an interesting problem. The application domains are very large.

**Q4 Main Weakness:**

The paper is difficult to read and does not allow to follow correctly the ideas. Experiments should be strengthened.

**Q5 Detailed Comments To The Authors:**

In addition, the different examples of proxy have to be compared and discussed, in particular with respect to the application (i.e. wrt why we know about the real data).
Please, discuss, for instance, the work of Lukusa (and coauthors) on this subject to complete the literature.

Lukusa, T. M., Lee, S. & Li, C. (2017). Review of Zero-Inflated Models with Missing Data. Current Research in Biostatistics, 7(1), 1-12. https://doi.org/10.3844/amjbsp.2017.1.12

Update: In the light of the discussion, I have raised my score (4) to a weak accept.

**Q9 Complying With Reviewing Instructions:**

Yes

---

> ### Author Rebuttal · Authors · 2024-04-05
>
> We thank the reviewer for their review and comments on our paper.
>
> The paper suggested by the reviewer is a great overview of prior work that considered missing data and zero inflation together, e.g., zero-inflation for response variable $Y$, while the covariate $X$ is potentially missing.  We would be happy to reference it, and a selection of key references contained there (in addition to our own prior work discussion).  We note that no prior work discussed in this paper, as far as we can see, considered the type of zero inflation we do, where a "?" or "NA" value in a missing data problem is replaced by a zero, meaning that excess zeros are produced by a systematic censoring process (rather than a systematic censoring process occurring in parallel with inflated zeros arising from a mixture model likelihood, as in many examples referenced in this paper).
>
> While most works mentioned in [1] consider MAR, more general types of missing data models could be represented using graphical models as done in [2,3,4]. Our motivation is to develop a similar graphical framework for zero-inflated data, thus enabling application of general graphical missing data methods to the type of zero-inflated data where some zeros are "real", and others replace true values, and are thus "fake".
>
> We would be happy to add additional experiments if the reviewer believes existing experiments are unconvincing, but we are not sure what the reviewer had in mind.
>
> [1] Lukusa, T. M., Lee, S. & Li, C. (2017). Review of Zero-Inflated Models with Missing Data. Current Research in Biostatistics, 7(1), 1-12. https://doi.org/10.3844/amjbsp.2017.1.12
>
> [2] Mohan, K., & Pearl, J. (2021). Graphical Models for Processing Missing Data. Journal of the American Statistical Association, 116(534), 1023–1037. https://doi.org/10.1080/01621459.2021.1874961
>
> [3] Malinsky, D., Shpitser, I., & Tchetgen Tchetgen, E. J. (2022). Semiparametric Inference for Nonmonotone Missing-Not-at-Random Data: The No Self-Censoring Model. Journal of the American Statistical Association, 117(539), 1415–1423. https://doi.org/10.1080/01621459.2020.1862669
>
> [4] Nabi, R., Bhattacharya, R., & Shpitser, I. (2020). Full Law Identification in Graphical Models of Missing Data: Completeness Results. Proceedings of the 37th International Conference on Machine Learning, 7153–7163. https://proceedings.mlr.press/v119/nabi20a.html

---

### Meta-Review · Area_Chair_rLYM · 2024-04-15

This paper studies the problem of modeling zero-inflated (ZI) data and proposes to approach it by considering it a missing data problem to attempt distinguishing true from technical zeros. The reviewers appreciate that the question is well motivated, and the approach makes sense, although the paper is not easy to follow.